# Triggering typical nemaline myopathy with compound heterozygous nebulin mutations reveals myofilament structural changes as pathomechanism

Johan Lindqvist[1], Weikang Ma [2], Frank Li[1], Yaeren Hernandez[1], Justin Kolb[1], Balazs Kiss[1], Paola Tonino[1], Robbert van der Pijl[1], Esmat Karimi[1], Henry Gong[2], Josh Strom[1], Zaynab Hourani[1], John E. Smith III [1], Coen Ottenheijm[1], Thomas Irving[2] & Henk Granzier [1,3 ✉]

Nebulin is a giant protein that winds around the actin filaments in the skeletal muscle sarcomere. Compound-heterozygous mutations in the nebulin gene (*NEB*) cause typical nemaline myopathy (NM), a muscle disorder characterized by muscle weakness with limited treatment options. We created a mouse model with a missense mutation p.Ser6366Ile and a deletion of *NEB* exon 55, the Compound-Het model that resembles typical NM. We show that Compound-Het mice are growth-retarded and have muscle weakness. Muscles have a reduced myofibrillar fractional-area and sarcomeres are disorganized, contain rod bodies, and have longer thin filaments. In contrast to nebulin-based severe NM where haplo-insufficiency is the disease driver, Compound-Het mice express normal amounts of nebulin. X-ray diffraction revealed that the actin filament is twisted with a larger radius, that tropomyosin and troponin behavior is altered, and that the myofilament spacing is increased. The unique disease mechanism of nebulin-based typical NM reveals novel therapeutic targets.

[1] Department of Cellular and Molecular Medicine, University of Arizona, 1656 East Mabel Street, Tucson, AZ 85724-5217, USA. [2] Department of Biology, Illinois Institute of Technology, Chicago, IL 60616, USA. [3] Sarver Molecular Cardiovascular Research Program, University of Arizona, Tucson, AZ 85721, USA. ✉email: granzier@email.arizona.edu

Nemaline myopathy (NM) is a heterogeneous disease with varying age of onset and severity and is characterized by skeletal muscle weakness, muscle atrophy, and intracellular nemaline rod bodies[1–5]. It can clinically be divided into subtypes of which the typical subtype (congenital and static or only slowly progressive) accounts for about half of the patients[5–8]. The most common cause of typical NM is mutations in the *NEB* gene that encodes the sarcomeric protein nebulin[3,9,10]. Nebulin is a ~800 kDa actin-binding protein that winds around the actin filament, with its C-terminus anchored in the Z-disk and its N-terminus near the thin filament pointed-end[11–18]. The structure of nebulin is highly repetitive with 35 amino acid modules that bind actin and that form super-repeats[11–14,19,20]. Knockout mice have revealed nebulin's importance for thin-filament length regulation and stability, force production (altered cross-bridge cycling kinetics and calcium sensitivity) and alignment of myofibrils[21–33].

Over 240 disease-causing mutations have been identified in *NEB*; the majority of patients are compound-heterozygous[9,34]. A missense mutation needs to be combined with a more disruptive mutation, such as a nonsense mutation, to cause typical NM[9]. The mechanisms by which mutations result in muscle weakness and atrophy are unclear. Current animal models with genetically modified nebulins fail to mimic typical NM patients due to, for example, their complete loss of nebulin and their dramatic phenotype that includes early mortality[21,22,27,35].

Here we created a mouse model that recapitulates the typical nebulin-based NM patient with compound-heterozygous mutations. The model carries a p.Ser6366Ile missense mutation in one actin-binding sequence in super repeat 18 (super-repeat 22 in the mouse) and a deletion of *Neb* exon 55. The missense mutation is a founder mutation with world-wide occurrence; in homozygous form it results in Finnish distal myopathy[36], but it has also been identified in combination with other *NEB* mutations resulting in compound-heterozygous NM[36–38]. In vitro studies with nebulin super-repeats harboring this p.Ser6366Ile mutation showed increased affinity for actin[39]. Deletion of *NEB* exon 55 was discovered as a founder mutation in Ashkenazi Jews, and has since been found to have global occurrence[40,41]. It results in intermediate or severe NM with diminished nebulin protein levels and muscle weakness[35,42]. By crossing the Ser6366Ile mouse model with our mouse model with *Neb* exon 55 deleted[35], a mouse was generated with compound-heterozygous *Neb* mutations. Functional, structural, and biochemical studies revealed altered thin filament structure, increased myofilament lattice spacing, a reduced myofibrillar fractional area, and reduced force production. This Compound-Het model will be useful for testing experimental therapies for typical NM.

## Results

### Creation and basic characterization of Compound-Het model.
A mouse model was created with a c.19097G>T mutation in the NEB gene that results in a serine to isoleucine substitution at the position corresponding to human Ser6366 in nebulin super-repeat 18 (or 22 in the mouse: Fig. 1a, and Supplementary Fig. 1), the Neb$^{S6366I}$ model. Mice from this line produce heterozygous, homozygous and wildtype (WT) mice according to Mendelian genetic ratios (Fig. 1b top). We bred heterozygous mice to heterozygous Neb$^{\Delta Exon55}$ mice, which produced the expected genotypes according to Mendelian genetic ratios (Fig. 1b bottom). Neb$^{S6366I/\Delta Exon55}$ and Neb$^{S6366I/S6366I}$ mice are referred to as Compound-Het and Neb$^{S6366I}$ Hom mice, respectively. Most studies used 4 months old mice, except when indicated otherwise. Compound-Het and Neb$^{S6366I}$ Hom animals have no outwardly visible phenotype with no unusual deaths during the timespan at

which they were studied (oldest mice in colony are 12 months old). Following mice over time revealed decreased body weights in the Compound-Het mice (Fig. 1c). Tibia lengths were slightly reduced (~2%) in both Compound-Het and Neb$^{S6366I}$ Hom mice (Fig. 1d). Grip strength was decreased in Compound-Het mice, by ~20% compared to WT mice (Fig. 1e). Neb$^{S6366I}$ Hom mice had a ~10% reduction at 10 mo (Fig. 1e). To study in situ muscle function, the force-frequency relation of the gastrocnemius muscle complex was measured with a hindlimb foot-plate system. In 3 mo old male mice of both genotypes forces were lower; the force reduction at the plateau of the force-frequency curve was 35% in Compound-Het mice and 7% in Neb$^{S6366I}$ Hom mice (Fig. 1f, left). In 10 mo old mice a similar force reduction occurred in Compound-Het mice (37%) and a significantly larger reduction in Neb$^{S6366I}$ Hom mice (18%) with identical results in female mice (Fig. 1f, middle and right). Normalized force-frequency curves in both the Compound-Het and Neb$^{S6366I}$ Hom mice were left-shifted, relative to WT (Fig. 1g). The stimulation frequency required for generating half-maximal force was decreased in Compound-Het mice (Fig. 1g, inset).

Skeletal muscle atrophy/hypotrophy is a common finding in NM[43,44], and muscle weights of the Compound-Het and Neb$^{S6366I}$ Hom were compared with WT littermates. Muscle weights in Compound-Het mice were mostly decreased, a weight of a few muscle types was unchanged or increased (Fig. 2a). In Neb$^{S6366I}$ Hom mice only two of the examined muscle types were changed: the gastrocnemius and plantaris had decreased weights (Fig. 2b). The decrease in plantaris weight was the same in the two genotypes (10%) but the gastrocnemius muscle weight loss was more pronounced in Compound-Het (20%) than Neb$^{S6366I}$ Hom mice (9%).

In summary, a Compound-Het mouse model (Neb$^{S6366I/\Delta Exon55}$) was created with reduced body weight, an in-situ force deficit and altered muscle trophicity.

### Compound-Het mice produce only Neb$^{S6366I}$ mutant protein.
To determine which type of mutant nebulin is expressed in the Compound-Het muscles (i.e., gene product of Neb$^{S6366I}$-allele and/or Neb$^{\Delta Exon55}$-allele), the Compound-Het mouse was crossed to a mouse with a smaller nebulin protein achieved by genetically removing three super-repeats, i.e., Neb$^{\Delta S9-11}$ (Kiss B. et al. in preparation). The smaller nebulin has a faster mobility during gel electrophoresis (deleting super-repeats 9-11 eliminates 729 amino acids) and this made it possible to identify the nebulin isoform in mice that contain the Neb$^{\Delta S9-11}$ allele and either the Neb$^{S6366I}$-allele or the Neb$^{\Delta Exon55}$-allele. By crossing the Neb$^{\Delta S9-11}$ mouse with the Neb$^{S6366I}$ mouse, a mouse is obtained that expresses a high mobility nebulin protein (produced by the Neb$^{\Delta S9-11}$-allele) and, if the Neb$^{S6366I}$-allele produces protein, a mutant protein with mobility similar to that of wildtype nebulin (the amino acid substitution in the Neb$^{S6366I}$ changes the molecular mass by 26 Dalton only). Results reveal that the Neb$^{S6366I}$-allele indeed produces nebulin protein (Fig. 3a). By crossing the Neb$^{\Delta S9-11}$-mouse with the Neb$^{\Delta Exon55}$-mouse a compound-heterozygous mouse was created with a high mobility nebulin protein (again produced by the Neb$^{\Delta S9-11}$-allele) and, if the Neb$^{\Delta Exon55}$-allele produces nebulin, a mutant protein with mobility similar to that of wildtype nebulin (exon 55 encodes 35 amino acid residues with a combined 4 kDa mass, a too-small size difference to cause a mobility shift on gels). Protein gels showed that none of the studied muscle types produce full-length nebulin at detectable levels, i.e., the Neb$^{\Delta Exon55}$-allele does not produce protein (Fig. 3a). Thus, it is likely that the nebulin protein produced by the Compound-Het mouse (Neb$^{S6366I/\Delta Exon55}$) is Neb$^{S6366I}$-nebulin.

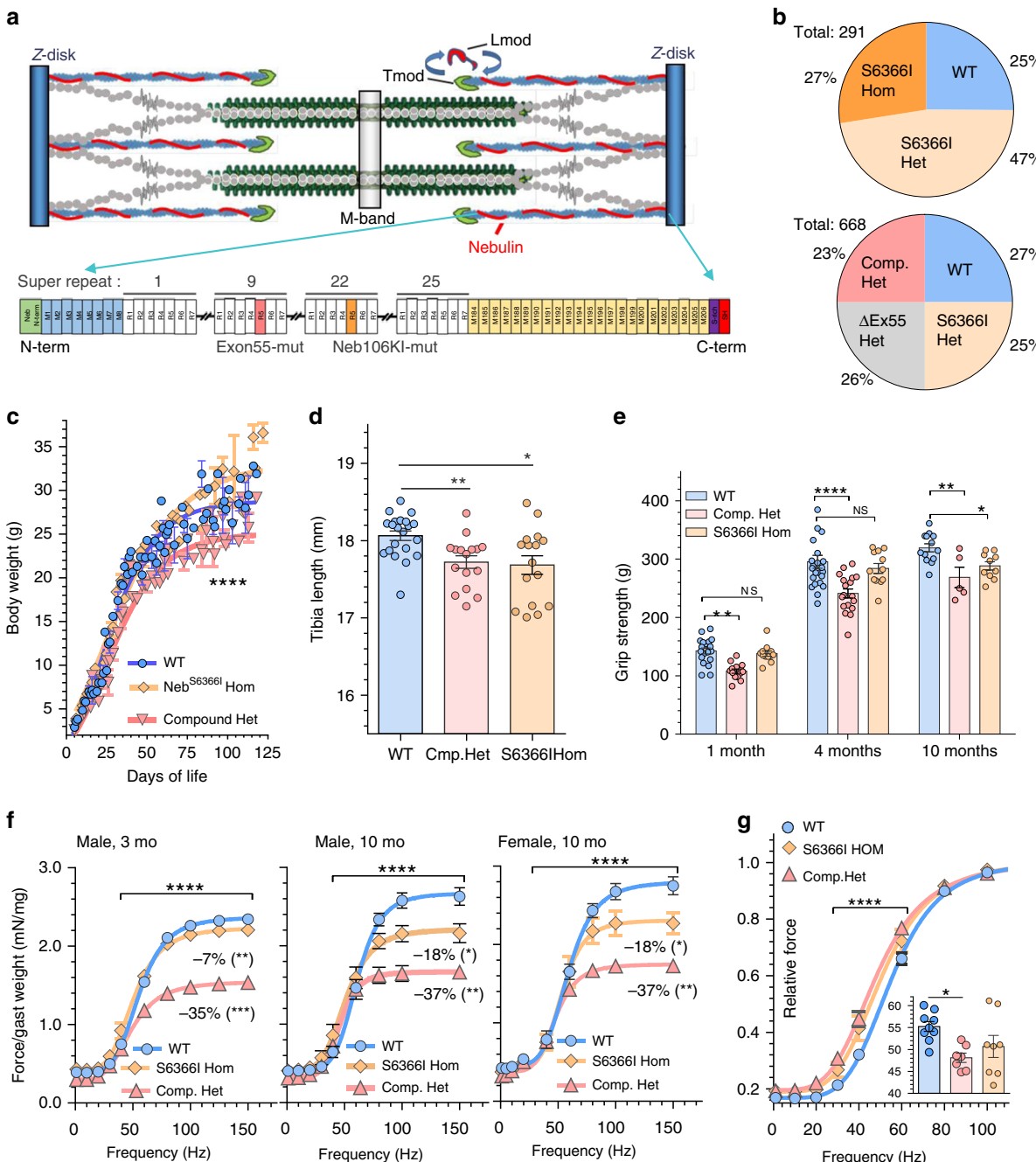

As other models of nebulin-based NM have decreased nebulin protein levels[21,22,27,35], we investigated the nebulin content in diaphragm and lower limb muscles in Compound-Het and Neb[S6366I] Hom animals. Nebulin protein levels were maintained at WT levels in all examined muscles from both genotypes (Fig. 3b, c). Thus, the Compound-Het mouse model (Neb[S6366I/ΔExon55]) is the first nebulin-based NM model that has normal nebulin protein levels.

**Changes in myosin heavy chain distribution.** A common finding in patients with nebulin-based NM and mouse models are changes in the proportion of myosin heavy chain (MHC) isoforms[27,29,44]. The Compound-Het mice (4 mo) followed this general trend and with the exception of the tibialis cranialis all examined muscles had changes in MHC expression (Fig. 4a). Soleus muscles had the most prominent change in type I myosin,

increasing from ~40% to ~70%. In most other muscles, type IIB myosin proportion decreased while IIA/X increased (Type IIA and IIX cannot be separated in our experiments and are quantified as a single band). Neb[S6366I] Hom animals, interestingly, displayed no changes in MHC proportion in any examined muscle type (Supplementary Fig. 2a). We also studied MHC isoform distribution in 2 mo and 10 mo old Compound-Het mice and focused on the EDL, soleus and TC muscles. Significant changes in the EDL and soleus muscle already existed in the 2 mo old mice and they persisted in the 10 mo old mice; a small shift in 2 mo and 10 mo old TC muscle was also found (Supplementary Fig. 2b). The same muscle types were also studied in 10 mo old S6366I mice which revealed in EDL and soleus muscle small but significant shifts toward MHC isoforms found in oxidative fibers (Supplementary Fig. 2c).

To gain additional insights into the fiber-type changes, cross-sections of EDL and soleus muscles were stained with antibodies

**Fig. 1 Basic characterization of the mouse strains. a** Top: Nebulin localizes to the thin filament of the skeletal muscle sarcomere. Bottom: The structure of mouse nebulin with the S6366I site indicated (in super-repeat 22, orange) and the ΔExon55 site (in super-repeat 9, pink). A mouse was created with compound heterozygous Neb mutations: one missense mutation in exon 106 (Neb[S6366I]) and one in-frame deletion of exon 55 (Neb[Δexon55]), the Neb[S6366I/Δexon55], or Compound-Het for short. **b** When breeding Het Neb[S6366I] parents, genotypes of offspring follow Mendelian genetic ratios (top) and when breeding Het Neb[S6366I] with Het Neb[Δexon55] parents (bottom), offspring also closely follows Mendelian genetics. **c** Body weight vs. days-of-life. Gompers non-linear least squares fit with as null hypothesis that one curve fits all data sets. Test (Sum-of-squares F-test) reveals that the null hypothesis is rejected ($p \leq 0.0001$; F (DFn, DFd): 63 (3,341)). (Results of male mice are shown with similar results found for female mice (not shown).) **d** Tibia length is reduced in Compound-Het and Neb[S6366I] homozygous 4 months old male mice. An ordinary one-way ANOVA without matching or paring was performed with a posthoc multiple testing comparison with multiple testing correction (Sidak). $*p = 0.02$; $**p = 0.008$. **e** Grip strength in Compound-Het and Neb[S6366I] mice, 1, 4 and 10 mo of age. A two-way ANOVA (no matching) reveals a significant effect of age and genotype on grip strength but without interaction. A posthoc multiple testing comparison with multiple testing correction (Dunnett) reveals a significant grip strength reduction at all 3 ages in Compound-Het mice and a significant reduction in 10 mo old Neb[S6366I] mice. $**p < 0.01$; $****p < 0.0001$. **f** In 3 mo old male mice (left) Compound-Het mice have a 35% decreased maximal lower limb force and Neb[S6366I] Hom have a 7% reduction. In 10 mo old male mice (middle) the reduction is 37% and 18% and in 10 mo old female mice (right) the reduction is 37% and 18%. Non-linear least squares fit to Hill curve with as null hypothesis that one curve fits all data sets. Test (Sum-of-squares F-test) reveals that the null hypothesis is rejected in each panel ($****p \leq 0.0001$; F (DFn, DFd): 104.2 (8,208)). **g** Relative force-frequency relation (same data as in f, left) shows a left shift for both the Compound-Het and Neb[S6366I] Hom mice with increased relative force at the shown frequencies of 40 and 60 Hz (One-way ANOVA without matching or paring performed with a posthoc multiple testing comparison with multiple testing correction (Sidak). $****p < 0.0001$. Inset: decreased frequency for ½ maximal force in Compound-Het mice. Ordinary one-way ANOVA without matching or paring. A posthoc multiple testing comparison with multiple testing correction (Sidak). $*p = 0.02$. Number of mice in **b** 291 (top) and 668 (bottom); **c, d**: WT: 21, Compound-Het: 18, and NebS6366I Hom: 16; **e** 1 mo 19, 14, 12; 4 mo: 22, 18, 12; 10 mo 14, 6, 10. **f** left and **g**: WT: 9, Compound-Het: 7, and Neb[S6366I] Hom: 6; **f** middle: WT: 6, Compound-Het: 2, and NebS6366I Hom: 6; **f** right WT: 6, Compound-Het: 4, and Neb[S6366I] Hom: 3. **g** Inset 9 WT mice, 6 Compound-Het mice and 7 Neb[S6366I] Hom mice. Bar values are mean ± SEM. Source data are provided as a Source Data file.

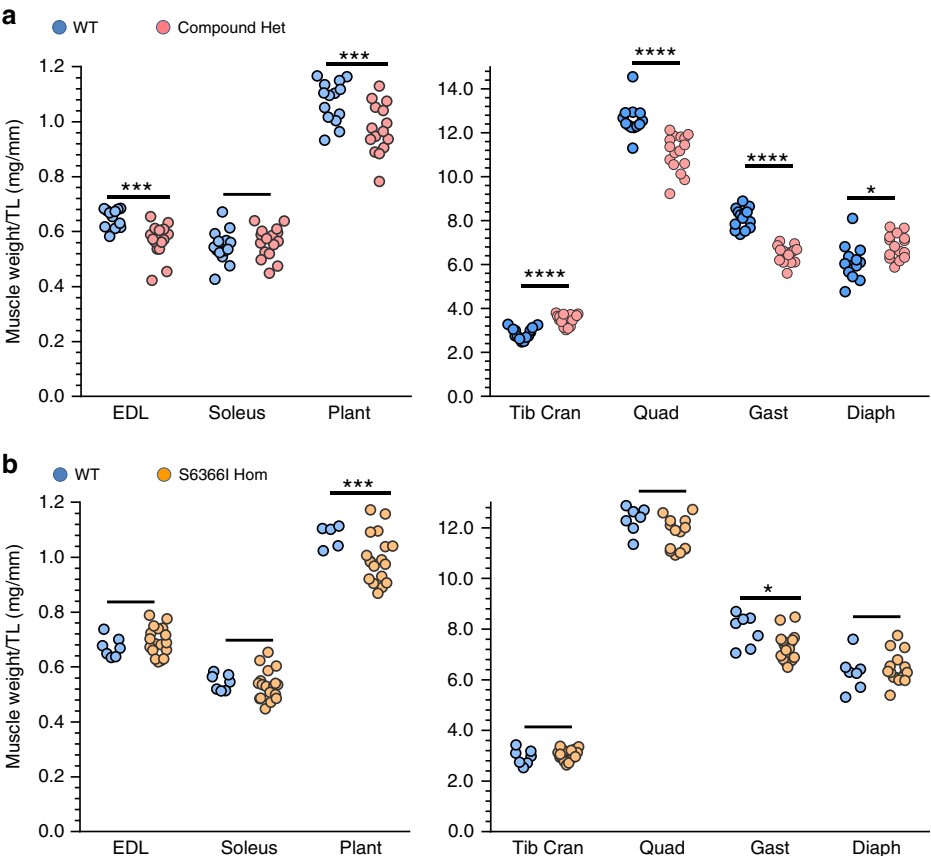

**Fig. 2 Assessing muscle weights. a** Compound-Het mice have decreased tissue weight in EDL, plantaris, quadriceps and gastrocnemius muscles while muscle weight of tibialis cranialis and diaphragm is increased. **b** Neb[S6366I] Hom mice have decreased muscle weight in plantaris and gastrocnemius. Number of mice in **a**: WT: 14, Compound Het: 16. Number of mice in **b**: WT: 7, and Neb[S6366I] Hom: 18. (Male mice (4 mo); similar findings in female mice, not shown; 2-Way ANOVA with multiple comparison (multiple comparison correction (Sidak)). $*p < 0.05$; $***p < 0.001$; $****p < 0.0001$). Source data are provided as a Source Data file.

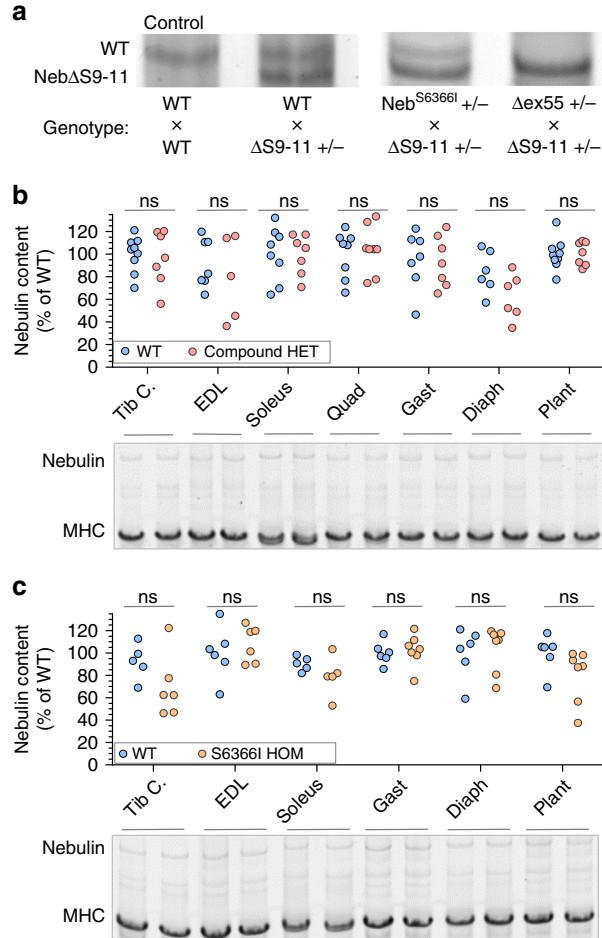

**Fig. 3 Nebulin protein expression. a** Nebulin isoform expression was assessed in muscles from mice with one Neb$^{\Delta S9-11}$-allele (this expresses a nebulin lacking super-repeats 9-11, generating a smaller than wild type nebulin) and the other allele either WT, Neb$^{S6366I}$ or Neb$^{\Delta ex55}$. The latter two alleles are expected to produce a nebulin protein of wildtype size (assuming they express protein, see text for details). Top: protein gels with nebulin protein bands; bottom: genotype of mouse from which muscle was obtained (gastrocnemius muscle). The left lane is from a WT mouse revealing WT nebulin and the second lane is from a heterozygous Neb$^{\Delta S9-11}$ mouse that reveals WT nebulin (top band) and a bottom nebulin band produced by the Neb$^{\Delta S9-11}$ allele. Lanes 3 and 4 show that the Neb$^{S6366I}$ allele produces full-length nebulin but that the Neb$^{\Delta ex55}$ does not. Identical results (not shown) were obtained in other examined muscle types (EDL, soleus Tibialis C. gastrocnemius and diaphragm), two mice per genotype with 6 muscle types studied per mouse. See text for details. **b, c** Expression levels of nebulin relative to myosin heavy chain (MHC) in a range of muscle types. Analyzed results (top) and gel examples (bottom) of **a** Compound-HET mice (Neb$^{S6366I/\Delta exon55}$), and **b** The Neb$^{S6366I}$ homozygous mice. No differences in nebulin protein levels were found in any of the studied muscle types. 2-Way ANOVA with multiple comparison (multiple comparison correction (Sidak)). ns not significant. Number of mice in **b**: WT: 9, Compound-Het: 8. Number of mice in **c**: WT: 6, and S6366I Hom: 7. Source data are provided as a Source Data file.

against the different myosin isoforms to determine fiber size and number (Fig. 4b, c left panels show examples). Because minimal changes were present in MHC expression in Neb$^{S6366I}$ Hom animals (see above), we focused this part of the work on only the Compound-Het mice. EDL from Compound-Het mice had a small reduction in the minFeret of type IIX fibers while the minFeret of the relatively few type I fibers doubled (Fig. 4b

middle). The size of type IIB fibers was unchanged but the number of type IIB fibers was decreased by ~20% in Compound-Het EDL muscles (Fig. 4b right). It is likely that the reduction in the number of type IIB fibers underlies the reduced EDL muscle mass. Soleus muscle from Compound-Het mice had a 50% decrease in type IIX minFeret and for IIA fibers minFeret trended down (Fig. 4c middle). The number of type I fibers was doubled while the number of type IIA fibers decreased by ~40% (Fig. 4c right). The large increase in the number of type I fibers is likely to explain the preserved soleus muscle mass in Compound-Het mice.

**Ultrastructural and histological effects of mutant nebulin.** Electron microscopy revealed nemaline rod bodies in EDL from both Compound-Het and Neb$^{S6366I}$ Hom mice that typically originated from the Z-disk (Fig. 5, yellow arrows). Interestingly, in soleus muscle rod bodies were only present in Compound-Het mice (Fig. 5a) and they are smaller and take up less fractional area than in EDL muscle (Fig. 5b, c). Wavy and misaligned Z-disks and areas with damaged Z-disks were also present in muscles from Compound-Het and Neb$^{S6366I}$ Hom mice (Fig. 5a). The presence of rod bodies has been examined in biopsies of Finnish distal myopathy patients that are homozygous for the Neb$^{S6366I}$ mutation[36]. (This mutation was initially designated p.Ser4665-Ile[36]). Only few and small nemaline rod bodies could be detected[36]. This earlier study on distal myopathy patients used histology with modified Gomori trichrome staining and we therefore also investigated rod bodies in gastrocnemius muscle of the mouse models, using the same method as used on patients. Gomori trichrome stained sections showed rod bodies in Compound-Het mice at both ages (Supplementary Fig. 3b, e). Furthermore, we detected small rod bodies in S6366I Hom mice at four and ten months of age (Supplementary Fig. 3c, f). These histology results in S6366I Hom mice are in overall agreement with those of the Finnish S6366I patients[36]. It is also noteworthy that core-like structures were observed in ten months old Compound-Het mice (Supplementary Fig. 3e). Cores have also been observed in myopathy patients with nebulin mutations[3,45].

Shorter thin filaments are frequently found in nebulin-based NM[46,47], and thin filament lengths were therefore measured on electron micrographs in Compound-Het and Neb$^{S6366I}$ Hom muscles. EDL muscles were studied because their H-zone can relatively easily be discerned, and thin filament lengths can be measured as the distance from the edge of the H-zone to the middle of the Z-disk (Supplementary Fig. 4a). Unlike what has been found in previous nebulin-based NM-studies, no decrease in thin filament lengths were detected. Instead a slight but significant increase in thin filament length was detected in both Compound-Het and Neb$^{S6366I}$ Hom muscles (Fig. 5d and Supplementary Fig. 4a–c). To validate this result, super-resolution optical microscopy was used to measure thin filament length in EDL muscle by labeling the thin-filament capping protein Tmod1[48]. This also revealed longer thin filaments in Compound-Het mice, compared to WT (Supplementary Fig. 4d). Thick filament lengths, measured by using the Ti102 antibody that labels the edge of the thick filament[49], revealed no differences (Fig. S4e).

**Influence of mutant nebulin on whole muscle function.** Muscle function of EDL and soleus was studied by electrically stimulating isolated intact muscles at a range of frequencies and measuring at the optimal length ($L_0$), isometric force. The maximal specific force produced by EDL muscle was reduced by ~22% in Compound-Het and ~17% in Neb$^{S6366I}$ Hom muscles (Fig. 6a inset). Normalizing the measured force at each frequency by the

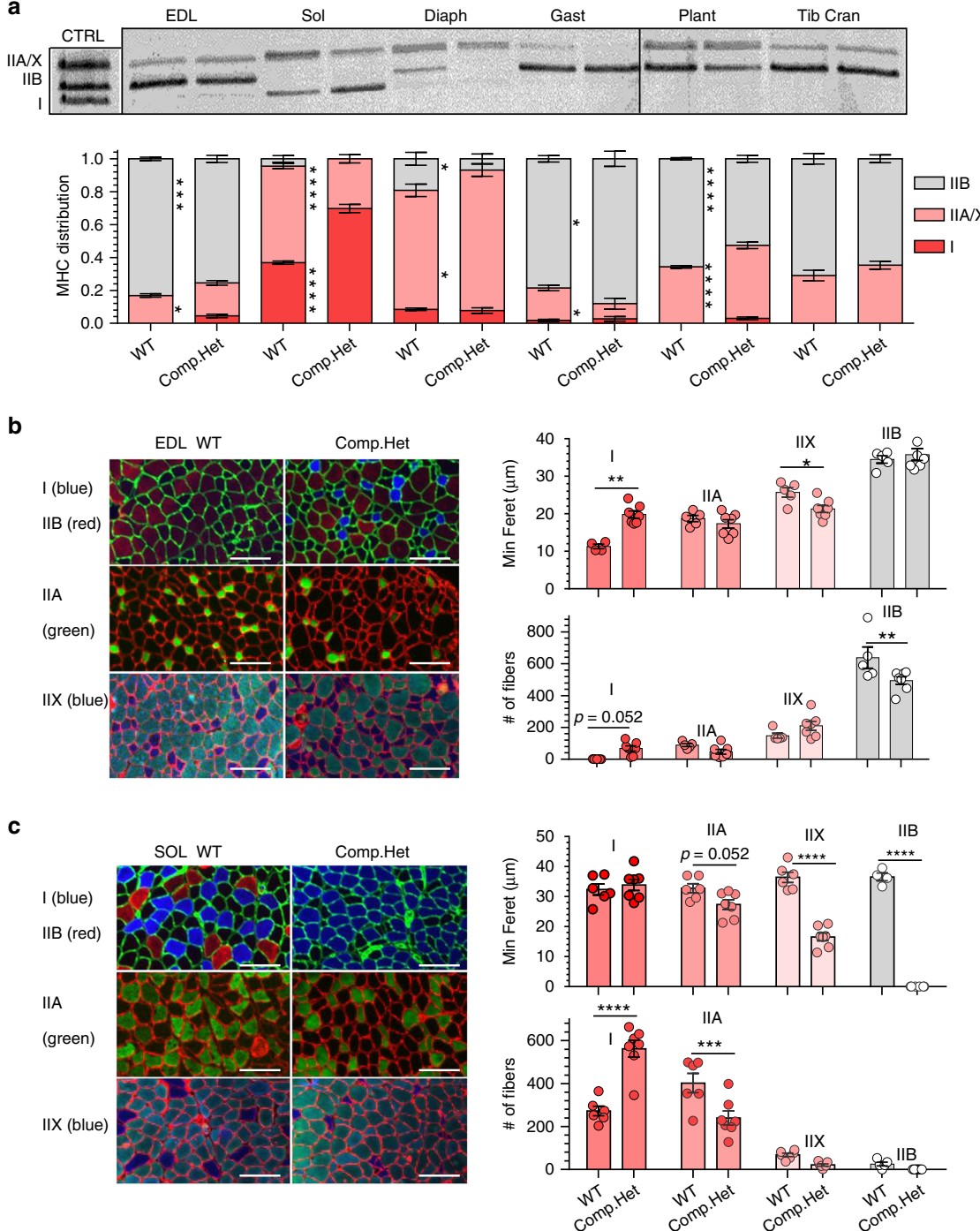

**Fig. 4 Myosin heavy chain (MHC) isoform distribution and CSA analysis. a** MHC expression in Compound-Het animals display significant changes in MHC distribution. A shift away from type IIB and toward slower myosin isoforms (either type IIA/X or type I) is observed in most muscle types (an exception is the tibialis m.). Top representative MHC gel images; bottom analyzed results. The control (CTRL) is a pooled homogenate of TC and Soleus muscle, revealing the 3 isoforms that can be discerned: type I, IIA/X and IIB (from bottom to top). **b** Example cross-sections stained with antibodies that are MHC-type specific (left), minFeret diameter (right, top) and number of fibers (right, bottom) in EDL muscle. MinFeret is increased in type I fibers and reduced in IIX. The number of type I fibers number is increased (only 3 type I fibers were found in WT mice) and the number of type IIB fibers is reduced. Scale bar: 100 μm. **c** Example cross-sections stained with antibodies that are MHC-type specific (left), minFeret diameter (right, top) and number of fibers (right, bottom) in soleus muscle. There is a reduction in type IIX and IIB minFeret and an increase in the number of type I and a reduced number of type IIA fibers. Scale bar: 100 μm. **a–c**: Two-way ANOVA (no matching) with a posthoc multiple testing comparison with multiple testing correction (Sidak). *$p < 0.05$, **$p < 0.01$; ***$p < 0.001$; ****$p < 0.0001$. Number of mice in **a**: WT: 9, Compound-Het: 7. Number of mice in **b** and **c**: WT: 6, Compound-Het: 7. Bar values are mean ± SEM. Source data are provided as a Source Data file.

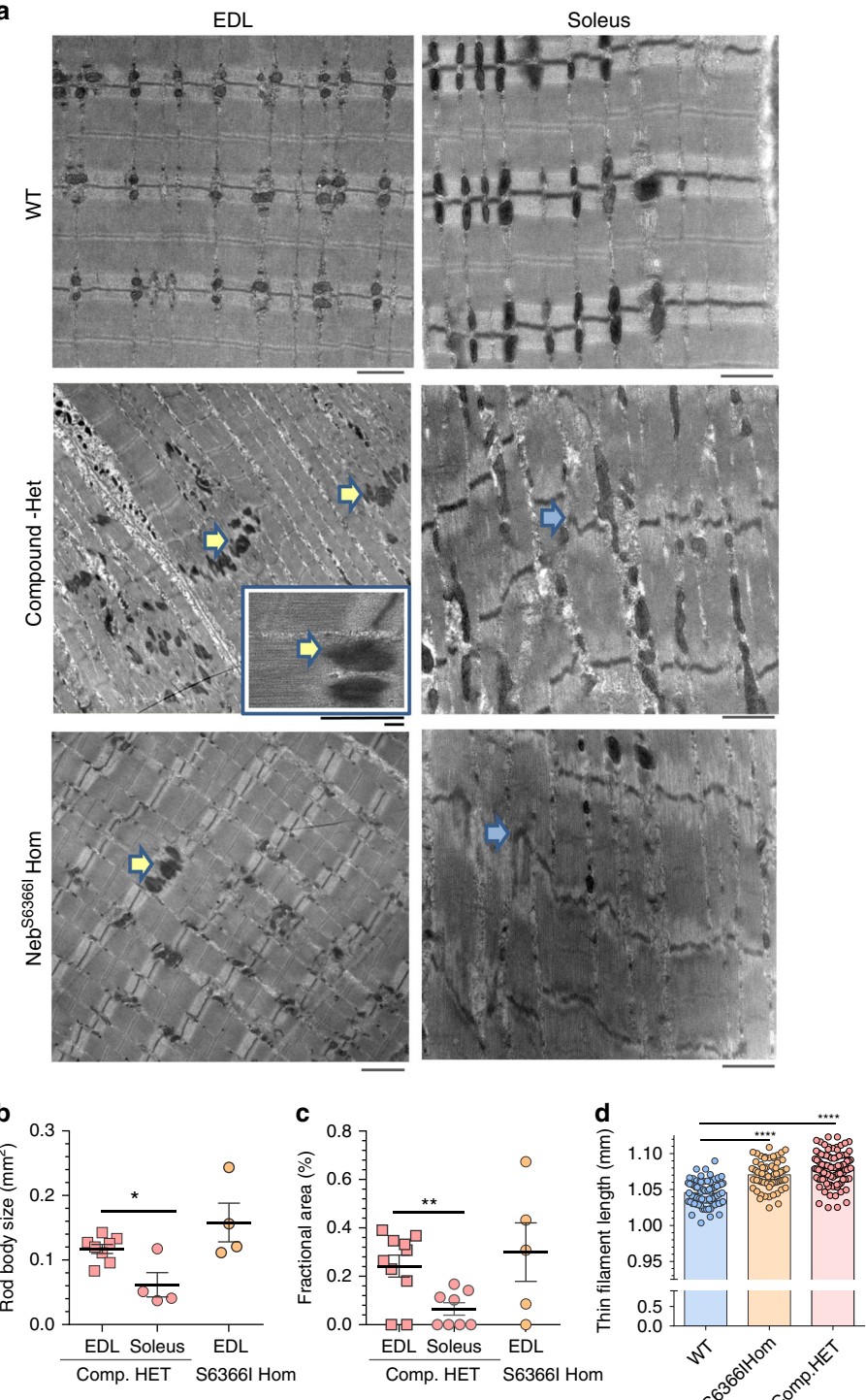

**Fig. 5 Nemaline rod bodies, sarcomeric ultrastructure, and thin filament length. a** Electron micrographs of intact EDL muscles (left) and soleus muscle (right) in WT (top), Compound-Het (middle) and Neb[S6336I] How (bottom). Nemaline rod bodies (yellow arrows) are found in both EDL and soleus of Compound-Het and in only EDL of Neb[S6366I] HOM mice. Inset in left middle panel shows enlarged rod bodies. In soleus muscle sarcomere disorganization was frequently seen in both Compound-HET and Neb[S6366I] HOM mice. Yellow arrows: rod bodies; blue arrows: altered/damaged Z-disks. Rod body size (**b**) and fractional area of rod bodies (**c**) in EDL and soleus. **b** *$p = 0.03$. **c** **$p = 0.0059$. Values are means ± SEM. Four mice per genotype were used with ~5 fibers per muscle examined. The mean value for each mouse is plotted. **d** Thin filament lengths are slightly but significantly longer in Compound Het and Neb[S6366I] Hom EDL muscle. Measurements made in SL range: 2.5–3.0 μm. (See Supplementary Fig. 4 for images and additional data, including confirmations by super-resolution optical microscopy.) **b–d** Ordinary one-way ANOVA without matching or paring. A posthoc multiple testing comparison with multiple testing correction (Tukey) was performed. **d** ****$p < 0.0001$. n(WT) = 81 measurements from 18 fibers from 6 mice; n(Hom) = 71 measurements from 18 fibers from 6 mice; $n$(Compound HET) = 103 measurements from 18 fibers from 6 mice. Horizontal lines are mean ± SEM. Source data are provided as a Source Data file.

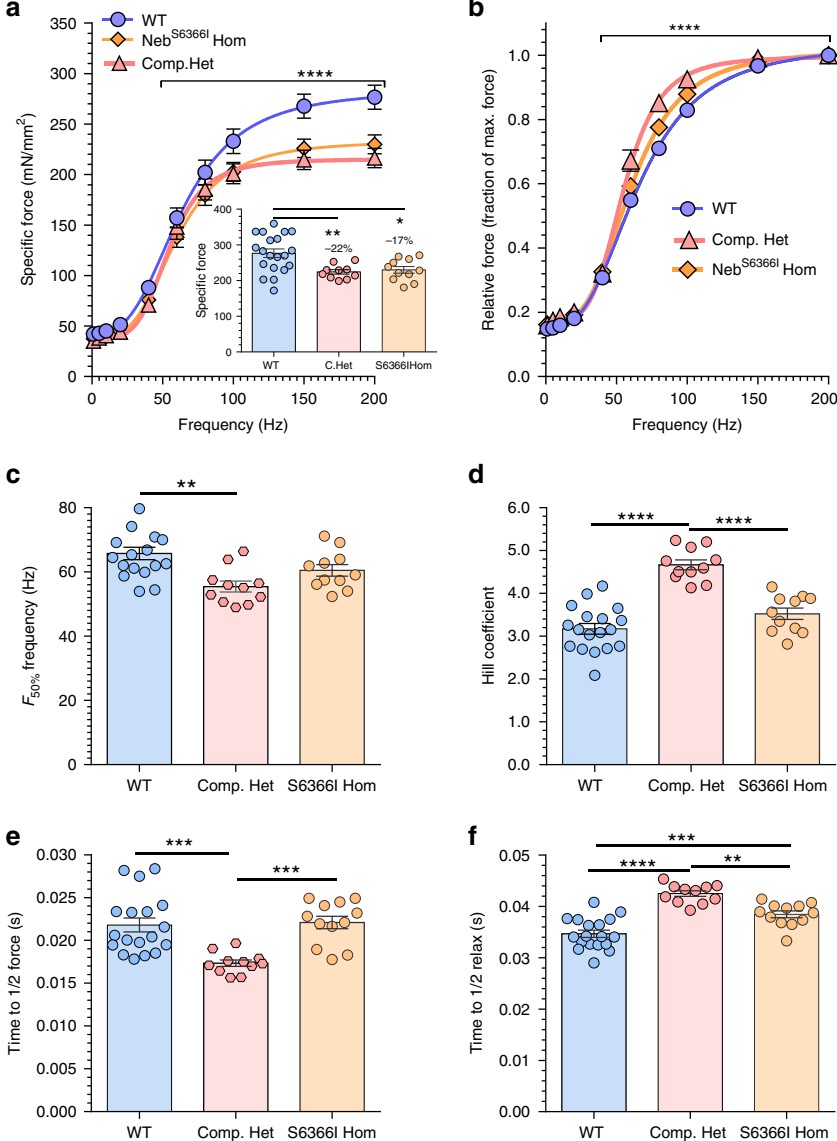

**Fig. 6 Muscle mechanics. a** Force-frequency relation of WT, Compound-Het and S6366I Hom EDL muscle. Inset shows maximal force (200 Hz stimulation rate). Force is reduced in Compound-Het and S6366I Hom mice. (Force expressed as specific force: force per unit area of muscle, mN/mm$^2$). *$p = 0.013$, **$p = 0.007$. **b** Relative force-frequency shows steeper force rise in Compound-Het mice. **c** Frequency for 50% force ($F_{50\%}$) is reduced in Compound-Het mice. **$p = 0.0012$. **d** Hill coefficient is increased in Compound-Het mice. ****$p < 0.0001$. **e** Time to 1/2 maximal force of 200 Hz tetanus is reduced in Compound-Het mice. ***$p = 0.0005$. **f** Time to 1/2 relaxation of 200 Hz tetanus is increased in Compound-Het mice and S6366I Hom mice. **$p = 0.002$, ***$p = 0.0008$ and ****$p < 0.0001$. **a–f**: $n(WT) = 20$ mice; $n(Compound-Het) = 11$ mice; $n(S6366I Hom) = 11$ mice. Bar values are mean ± SEM. **a**, **b** Non-linear least squares fit to Hill curve with as null hypothesis that one curve fits all data sets. Test (Sum-of-squares F-test) reveals that the null hypothesis is rejected in each panel ($p \leq 0.0001$, F (DFn, DFd) 9.633 (8,408)). **a** inset and **c–f**: ordinary one-way ANOVA without matching or paring. A posthoc multiple testing comparison with multiple testing correction (Tukey) was performed. Source data are provided as a Source Data file.

maximal force (200 Hz) revealed for the Compound-Het muscles a force increase at sub-maximal stimulations with a corresponding decrease in the frequency for half-maximal force, and increased cooperativity (Fig. 6b–d). EDL from Compound-Het mice took a shorter time to produce half-maximal force during tetanic stimulation (Fig. 6e) but longer time to half relaxation (Fig. 6f). Neb$^{S6366I}$ Hom EDL muscles also relaxed more slowly, as indicated by the longer time for half relaxation, but not as long as for Compound-Het muscles (Fig. 6f). Soleus muscles had preserved specific force in Compound-Het and Neb$^{S6366I}$ Hom muscles (Supplementary Fig. 5a). The Compound-Het soleus muscle had a reduced frequency for half-maximal activation, and an increased Hill-coefficient and slowed relaxation time (Supplementary Fig. 5b–d, f). The Neb$^{S6366I}$ Hom soleus muscle

only had slowed relaxation times (Supplementary Fig. 5f). In summary, EDL muscle from both mouse models had impaired force production and slowed relaxation whereas the phenotype in soleus muscle was much milder. The mechanistic basis of these finding was studied next, using single fiber mechanics and X-ray diffraction on EDL muscles from Compound-Het mice, where the contractility phenotype was most severe.

**Changes in single fiber force and calcium sensitivity**. To establish whether the depressed contractility of intact muscle in Compound-Het mice has a myofilament-basis, single fiber studies were conducted. Membrane-permeabilized fibers from EDL muscles were used and force-pCa-curves (pCa = −log([Ca$^{2+}$]),

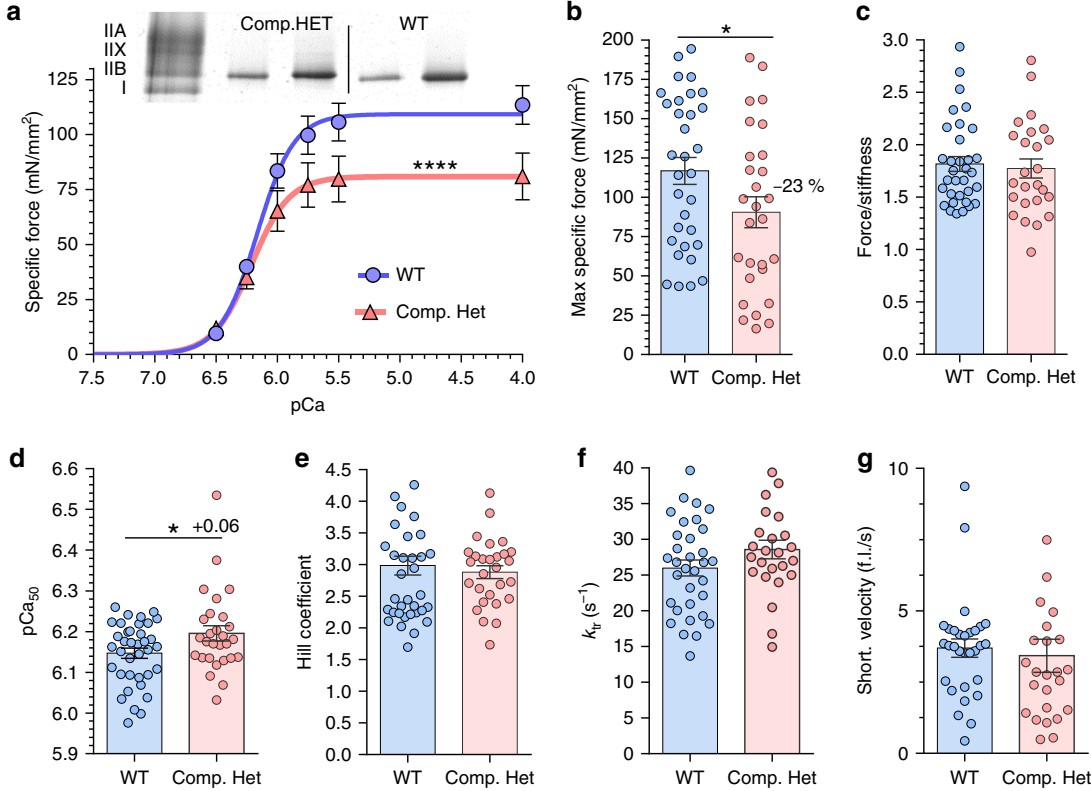

**Fig. 7 Single fiber mechanics in type IIB fibers (EDL muscle) of Compound-Het mice. a** Top: MHC-gel. Examples of IIB single fibers that were used in experiment. Two Compound-Het and two WT fibers are shown and a total of 41 WT and 40 Compound-Het fibers were studied; (WT: 34/41 fibers were IIB; Compound-Het: 28/40 fibers were IIB). The left lane is a standard with all MHC isoforms. **a** Bottom: Specific force –pCa relations of Compound-Het and WT fibers. A Hill-type curve-fit analysis shows that the Compound-Het fibers are significantly different from WT fibers. **b** Maximal specific force is reduced in Compound-Het fibers ($p = 0.046$). **c** The force/stiffness relation is unaltered. Increase in calcium sensitivity ($p = 0.028$) (**d**) but unchanged cooperativity (**e**) in Compound-Het fibers. (Values based on force-pCa curves that were normalized to the maximal specific force.) Kinetics of tension redevelopment as reflected in $k_{tr}$ (**f**) and maximal shortening velocity (**g**) are unaltered in Compound-Het fibers. The data set comprises on average 7 type IIB fibers from each of 5 WT mice and 5 type IIB fibers from each of 6 Compound-HET mice. Values are mean ± SEM. **a**: allosteric sigmoidal non-linear least squares fit with as null hypothesis that one curve fits all data sets. Test (Sum-of-squares F-test) reveals that the null hypothesis is rejected ($p \leq 0.0001$, F (DFn, DFd) 8.146 (3,360)). **b–g** two-tailed unpaired t-test. *$p < 0.05$; ****$p < 0.0001$. Source data are provided as a Source Data file.

active stiffness, rate constant of force redevelopment ($k_{tr}$) and maximum unloaded shortening velocity were measured at a sarcomere length of 2.5 μm. Afterward, the fiber was recovered, and its myosin isoform was determined. As fibers from WT and Compound-Het mice were mostly type IIB, analysis was restricted to this fiber type (Fig. 7a, top). We observed a 23% reduction in maximal specific-force generated by Compound-Het fibers (Fig. 7a, bottom and Fig. 7b). Active stiffness is a measurement of the number of force-generating cross-bridges and series compliance. No change in stiffness or the force/stiffness ratio was observed when comparing Compound-Het and control fibers (Fig. 7c). Interestingly, a small leftward shift in the force-pCa relationship was detected in Compound-Het fibers due to increased calcium sensitivity while cooperativity was unaffected (Fig. 7d, e). The $k_{tr}$ was determined by rapidly shortening and re-stretching of the fiber and fitting an exponential curve to the force redevelopment curve. $k_{tr}$ was not different between genotypes (Fig. 7f). Finally, the unloaded shortening velocity, measured by performing the slack test procedure[50], was maintained in fibers from Compound-Het mice (Fig. 7g). Thus, Compound-Het fibers have a reduced maximal specific-force whereas calcium sensitivity is increased and the kinetic parameters are unaltered.

**X-ray diffraction reveals thin filament structural alterations.** Small-angle X-ray diffraction is a powerful technique to study the

structural alterations caused by mutations in myofilament proteins, under physiological conditions in intact muscle. The EDL muscle was studied in its passive state before activation, and during the plateau of a tetanic contraction in Compound-Het and WT mice. A schematic of the setup and typical X-ray diffraction images are in Fig. 8a, b. We focused on thin filament-based reflections and first determined the stiffness of the thin filaments, using the 27 Å meridional reflection that arises from the actin subunit spacing. The spacing of the 27 Å reflection during the tetanic force plateau was increased by ~0.3% in both WT and Compound-Het muscles (Fig. 8c). The stiffness of thin filaments was determined from the 27 Å spacing change and by converting the tetanic force of the whole muscle to force per thin filament[30], as explained in Supplementary Fig. 6. Stiffness values so obtained were identical in WT and Compound-Het muscles, 33.4 ± 3.5 and 33.6 ± 2.4 pN/nm/μm (Fig. 8d). Unlike in nebulin deficient muscle where the thin filament stiffness is 3-fold reduced[30], thin filament stiffness is unaltered in the Compound-Het model.

The thin filament helix was also studied, by analyzing the 6th and the 7th actin layer lines (ALL6 and ALL7), which occur at ~59 Å and ~51 Å, respectively. The 51 Å spacing was unchanged (Fig. 8e, left) but, interestingly, the 59 Å spacing was significantly reduced (Fig. 8e, right). The ALL6 also had a reduced intensity (see Supplementary Fig. 6c). These findings suggest that the thin filament helix is twisted relative to WT in the Compound-Het.

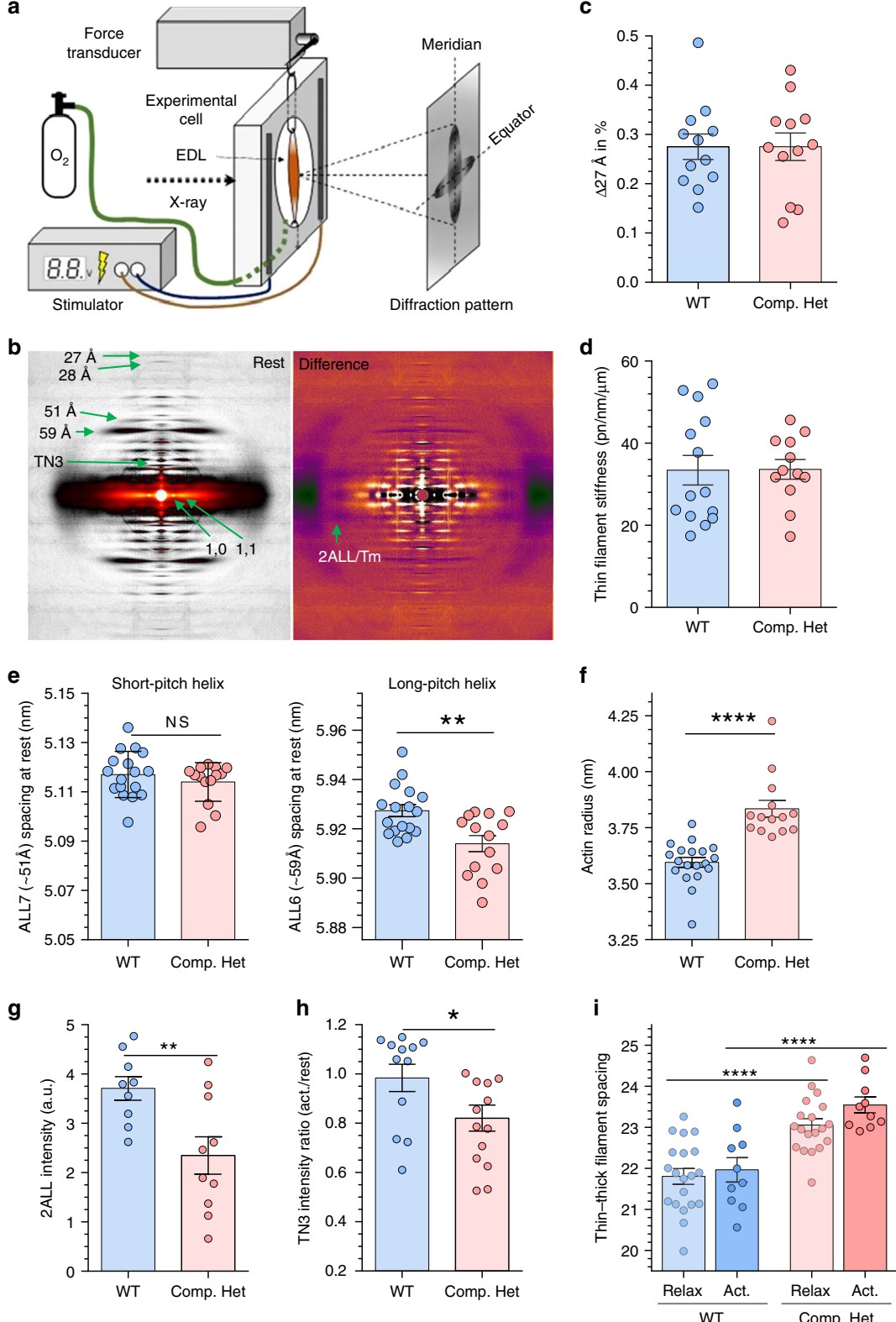

To determine if this twist has an effect on the actin radius, the radial separation of the intensity maxima of ALL6 was measured and observed to be significantly decreased in the Compound-Het, indicating an increased actin filament radius in the Compound-Het (Fig. 8f).

To determine whether the alteration in the long-pitch actin helix affects tropomyosin and troponin behavior, the off-meridional second actin layer line (2ALL (tropomyosin)) and the meridional third-order troponin (TN3) reflections were analyzed. Significantly less intense 2ALL and TN3 reflections during tetanic contraction were found in the Compound-Het EDL (Fig. 8g, h). Thus, it appears that structural changes in the actin helix caused by the nebulin mutation also alter the behavior of troponin and tropomyosin during contraction.

**Fig. 8 Thin filament structure assessed by X-ray diffraction on intact EDL muscle. a** Experimental setup. **b** X-ray diffraction images with examined reflections indicated. Left image taken during rest and Right image is the difference between active and passive muscle. (See Methods for details.) **c** The spacing change of the 13th actin meridional reflection (27 Å), during tetanic force development, is identical in both genotypes. **d** Thin filament stiffness is identical in both genotypes (for calculations, see Supplementary Fig. 6b, c and Results). **e** Short(left) - and long(right)-pitch helical periodicity of the actin filament. The long-pitch helix (right) is significantly reduced ($p = 0.0025$) in the Compound-Het. **f** The actin radius, determined from the radial spacing of the ALL6, is increased in Compound-Het. **g** The intensity change of the second actin layer line (due to tropomyosin) and **h** the intensity of the third meridional troponin reflection upon activation, are both reduced in the Compound-Het EDL muscle ($P = 0.0089$ and $p = 0.044$., respectively). **i** Thin-thick filament spacing ($2/3 \times d1,0$) is increased in compound Het, both in the relaxed and active states. (Based on d1,0 data in Supplementary Fig. 6a) Number of mice: WT: 15, Compound-HET: 11. For some animals both EDL muscles were used. Number of muscles: WT: 18, Compound-HET: 14. Bars and lines are mean ± SEM. **c–h** two-tailed unpaired t-test. *$p < 0.05$;**$p < 0.01$; ****$p < 0.0001$. **h** Ordinary one-way ANOVA without matching or paring. A posthoc multiple testing comparison with multiple testing correction (Tukey) was performed. ****$p < 0.0001$. Source data are provided as a Source Data file.

Finally, the increased actin radius in Compound-Het muscles might affect myofilament lattice spacing, therefore we determined the thin-thick filament distances. This revealed that the distance between thin and thick filaments were ~1.4 nm larger in both passive and active Compound-Het muscles (Fig. 8i).

## Discussion

Mutations in nebulin are the most common genetic cause of NM, and most often present as typical NM, i.e., congenital muscle weakness that is not or only slowly progressive with a relatively normal life span[3]. Most disease-causing *NEB* mutations are compound-heterozygous with a missense mutation on one *NEB*-allele and a more disruptive mutation on the other[3,4,9]. Many of the existing nebulin-based mouse models do not mimic mutations in patients but instead were created to study the basic functions of nebulin[21,22,27,29,51]. An exception is the Neb$^{\Delta ex55}$ model[35] that mimics NM patients with an in-frame deletion of exon 55[41]. However, homozygous Neb$^{\Delta ex55}$ mice have a more severe phenotype than Neb$^{\Delta ex55}$ patients, as shown by their survival of less than a few days and near-complete disappearance of nebulin protein[35], whereas homozygous Neb$^{\Delta ex55}$ patients survive to adulthood and express 20–50% of control levels of nebulin protein[9,42]. Thus, existing mouse models do not represent nebulin-based typical NM. Here we successfully created a compound-heterozygous mouse that carries the *Neb* exon 55 deletion and the p.Ser6366Ile missense mutation, both identified as a myopathic founder mutation with world-wide occurrence[36–38].

Existing nebulin models and patients with severe NM have reduced nebulin expression and shorter thin filaments[21,22,26,27,42], a reduced $k_{tr}$ and lower calcium sensitivity[23,27,42,46], and reduced force per crossbridge[31]. None of these changes are present in the Compound-Het where nebulin expression is unaltered (Fig. 3b), thin-filament length is slightly longer (Fig. 5d, Supplementary Fig. 4d), $k_{tr}$ is normal and calcium sensitivity is increased (Fig. 7d, f). Despite the normal levels of nebulin, NM characteristics are clearly present (e.g., nemaline rod bodies, muscle weakness, muscle atrophy, etc.). Additionally, a fiber-type shift toward slower myosin isoforms occurs in Compound-Hets (Fig. 4), as seen in patients with NM and existing nebulin mouse models[8,27,52]. Hence, factors other than nebulin content per se can cause nebulin-based NM, and typical and severe NM have different disease mechanisms.

The nebulin protein expressed in the Compound-Het model could be transcribed from either *Neb*-allele. Crossing the Compound-Het or the Neb$^{S6366I}$ to a model that expresses a smaller nebulin (Neb$^{\Delta S9-11}$) provides evidence that the Neb-$^{\Delta Exon55}$-allele does not produce protein and that the Neb$^{S6366I}$-allele does produce nebulin (Fig. 3a). It is therefore likely that the Compound-Het mice express mainly nebulin with the S6366I substitution. The Compound-Het mouse has a more severe phenotype than the Neb$^{S6366I}$ Hom model (e.g., Figs. 1e, f, 2, 5,

and 6) even though both models are likely to express exclusively nebulin with the S63661I substitution and, furthermore, both models express nebulin at normal WT levels (Fig. 3b, c). This indicates that the Compound-Het is somehow affected by the Neb$^{\Delta Exon55}$-allele. This could be due to a minute level of protein that is expressed by the Neb$^{\Delta Exon55}$-allele but that escapes detection in the protein gel assays. For example, ΔExon55-nebulin might not be stable and degrade easily and, therefore, not show up as a full-length protein, with some of the degradation product functioning as a poison-peptide. In addition, the loss of Neb$^{\Delta Exon55}$ transcript, possibly due to mRNA-mediated decay[35], could cause cellular stress that exacerbates the phenotype in Compound-Het mice. An alternative explanation for the more severe phenotype of the Compound-Het is that a single Neb$^{S6366I}$-allele is unable to supply enough nebulin to accommodate the normal rate of protein turnover. Consequently, the S6366I protein might have a longer half-life in the Compound-Het than in the Neb$^{S6366I}$ Hom, and perhaps accumulate additional post-translational modifications that are deleterious. Understanding how a single allele with a missense mutation produces a more severe myopathy compared to a homozygous state with the missense mutations warrants future studies. Considering that typical NM patients often are compound heterozygous with one nonsense and one missense *NEB* mutation[3,4,9], these future studies have high clinical relevance.

Nebulin comprises 25 super-repeats, each containing 7 SDxxYK actin-binding motifs[18]. The super-repeats at the ends of the molecule were recently shown to bind several-fold more strongly to actin in vitro than the central super-repeats[53]. The p. Ser6366Ile mutation is found in super-repeat 18 (or 22 in the mouse) which supposedly binds weakly to actin. Interestingly, the Ser6366Ile mutation strengthens the actin-binding affinity of nebulin[39]. The muscle weakness of both the Compound-Het and Neb$^{S6366I}$ Hom models suggests that it is important to have weak-binding super-repeats in the central region of nebulin.

Initially we considered the possibility that the S6366I mutation alters thin filament stiffness and X-ray diffraction studies were conducted to measure the extension of the actin subunit spacing (27 Å reflection) during activation. By combining this measurement with the measured tetanic force, scaled down to force per thin filament, a thin filament stiffness value of 33.6 pN/nm/μm in Compound-Het muscle is obtained which is indistinguishable from the 33.4 pN/nm/μm in WT muscle. Thus, it is unlikely that altered thin filament stiffness plays a role in the phenotype of the Compound-Het mouse.

An earlier image reconstruction study reported evidence for multiple nebulin binding sites on the outer domains of each actin subunit[54]. Furthermore, it has been shown that the presence of nebulin facilitates tropomyosin movement during tetanic contraction[30]. These studies support the notion that during contraction, nebulin moves on the actin filament, in conjunction with tropomyosin. It is conceivable that such movement requires weak

binding between nebulin and actin and, thus, an increase in actin-binding affinity (Ser6366Ile mutation) might be deleterious.

The X-ray diffraction study revealed changes in the F-actin structure in the Compound-Het as shown by the significantly reduced ~59 Å spacing and unaltered ~51 Å spacing (Fig. 8e). The spacings of the 59 Å and 51 Å layer lines correspond to the pitches of the left-handed and right-handed one-start helices of the F-actin filament, respectively (e.g., see Fig. 2a of Volkmann et al.[55]). These 59 Å and 51 Å layer line spacings are known to slightly increase during isometric contraction and decrease during unloaded shortening, reflecting changes in thin filament strain[56]. However, they do not behave identically, during neither isometric contraction nor unloaded shortening, reflecting changes in thin filament twist[56]. Importantly, we previously found in the nebulin cKO mouse that the 59 Å spacing is reduced without affecting the 51 Å spacing nor the baseline actin monomer axial spacing (27 Å), revealing that nebulin increases the thin filament helical twist[30]. The findings of the present work in the Compound-Het model (unaltered 51 Å spacing, reduced 59 Å spacing, no change in actin monomer spacing) are consistent with this observation and support that nebulin plays a role in the twist of the F-actin filament. Our finding that the thin filament diameter is slightly but significantly increased (Fig. 8f) likely reflects, at least in part, the increased helical twist. Bordas et al.[56] developed an analytical expression for calculating the degree of twist and its direction using the differential spacing changes of the 51 Å and 59 Å reflections (for details, see[56]). This approach was used here (Fig. 8e) to calculate in Compound-Het muscle the degree of F-actin helical twist, defined in terms of $\Delta n/n$ where n is the number of subunits per turn of the actin double helix. A 0.9% increase in actin filament twist was found in the Compound-Het EDL muscles as compared to WT. This is accomplished with no significant change in the 27 Å axial repeat of the subunits (Fig. 8c) consistent with a pure rotation of the subunits around the long axis.

Although the changes in the structure of F-actin in Compound-Het mice are small they nevertheless might have important consequences for thin filament activation and acto-myosin interaction. Consistent with this possibility, we found altered tropomyosin and troponin behavior in the Compound-Het, as indicated by the reduced intensity of the 2ALL and the TN3 reflections during contraction (Fig. 8g, h). This might underlie the altered calcium sensitivity of skinned fibers (Fig. 7d) and contribute to the left-shift in the force-frequency relation of intact muscle (Figs. 1g, 6b, c). These changes, however, are unlikely to cause disease, as they will increase force at sub-maximal stimulation rates. It is worth pointing out, however, that increased calcium sensitivity is also expected to slow down the speed of relaxation, as was found in intact EDL and soleus muscle (Fig. 6f, Supplementary Fig. 5f), and that this might result in loss of functionality during rapid movements.

In addition, the X-ray diffraction studies revealed a ~1.4 nm larger spacing between the thin and thick filaments (Fig. 8i) which is expected to decrease the number of myofilaments per unit cross-sectional area of myofibril by 12.9% (Supplementary Fig. 6b) and, hence, decrease specific force by 12.9%. Considering the constant volume behavior of intact skeletal muscle[57] the increased lattice spacing of the Compound-Het muscle could, in principle, be due to a 12.9% shorter sarcomere length. Sarcomere length was measured in passive muscle at $L_0$ in a subset of the muscles used for X-ray diffraction (4 WT and 6 Compound-Het muscles) and the values obtained were indistinguishable (2.76 ± 0.08 μm in WT and 2.69 ± 0.04 μm in Compound-Het; $p = 0.41$). Furthermore, thin filament length was slightly increased in Compound-Het (Fig. 5d), indicating no underlying structural cause for decreased sarcomere length in Compound-Het muscles.

Could the expanded myofilament lattice be due to the structural effect of the Ser6366Ile mutation on the thin filament? The factors that determine lattice spacing are complex, and include thin and thick filament charges that cause electrostatic repulsive forces between filaments as well as lateral forces due to titin[58], cross-bridges and Z-disk and M-band structures[57]. We speculate that the twist and increased radius of the actin filament in the Ser6366Ile mouse result in altered charge distributions on the thin filament and that this contributes to the lattice expansion. Due to the complexity of the factors involved in setting the myofilament lattice spacing, future studies focused on the myofilament lattice are warranted.

A second factor that will affect the specific force of EDL muscle in the Compound-Het is the reduced fraction of the fiber cross-section that is taken up by myofibrils, 0.766 in Compound-Het vs 0.801 in WT ($p < 0.01$). This is expected to lower force by ~3.5%. This reduced fractional area is explained by the slightly higher fractional area of mitochondria in the Compound-Het muscles (Supplementary Fig. 6b). The reduced myofibrillar area and the above discussed myofilament lattice expansion add up to an expected specific-force deficit of 16.4%, a considerable portion of the 23% force deficit in EDL fibers. In addition, wavy, misaligned and damaged Z-disks were found during ultrastructural examination of Compound-Hets (Fig. 5), which likely contributes to the muscle weakness. Finally, atrophy (13% in EDL muscle) will reduce the total cross-sectional area of muscle which is expected to further lower the total force produced, in addition to the reduced specific force.

To summarize, we created and extensively characterized a typical NM mouse model with compound heterozygous mutations that mimic those found in NM patients. Compound-Het mice are growth-retarded, have atrophic muscles that contain nemaline rods, and have muscle weakness. Structural studies show that the thin filaments are slightly twisted and compared to wildtype thin filaments, tropomyosin and troponin behavior during activation is altered. Furthermore, the myofilament lattice spacing is increased and the fractional area of myofibrils is reduced. The Compound-Het myopathy takes place while nebulin expression is normal, thin filament lengths are slightly increased and crossbridge kinetics are unaltered. This contrasts with nebulin-based severe NM (e.g., Neb$^{\Delta ex55}$) where nebulin expression is severely reduced, thin filaments are shorter, calcium sensitivity is reduced, and the force per cross-bridge and the fraction of force-generating cross-bridges are reduced, in mice[21,27,31] and patients[42,46,47]. Thus, the present study highlights the importance of understanding the different disease mechanisms that cause typical and severe NM, and that it is not possible to extrapolate from one to the other. Future studies are required to establish whether patho-mechanisms at play in patients with typical NM are similar to those in the Compound-Het mouse model. The Neb$^{S6366I}$ Hom model has a phenotype that develops with age and appears to share similarities with Finnish distal myopathy patients[36]. The mechanistic correspondence between this model and patients also warrants further research. The models that were developed in this work will be useful for deciphering the pathophysiological mechanisms of typical NM and Finnish distal myopathy and for developing therapeutic approaches, to increase muscle strength, and bring much needed relief.

## Methods

**Generation of Compound-Het mouse model.** The generation of Neb$^{\Delta Exon55}$ has been described elsewhere[35]. To create the Neb$^{S6366I}$-mouse, a targeting vector to Knock-In the Ser -> Ile missense mutation was constructed with the Osdd vector which allows thymidine kinase as a negative selection cassette. (Note that this mutation was initially designated p.Ser4665Ile, but has been re-designated

p.Ser6366Ile (www.lovd.nl/NEB.).) Overlapping primers introduced two nucleotide changes, one change created an MfeI restriction site (to facilitate distinguishing the targeted Knock-in and endogenous transcripts as well as allowing design of allele-specific PCR primers) and the other change resulted in the Ser -> Ile missense mutation corresponding to c.19097G>T found in humans[9,36]. Linearized targeting vector was electroporated into 129S6/SvEvTac cells; 6 G418-resistant ES cell clones were confirmed to have correctly integrated into the Neb gene. The clones were sequenced to verify the introduction of the mutation. The cells were injected into C57BL/6J blastocysts before transfer to recipient hosts. Chimeras were bred with C57BL/6J mice and neo-containing mice were identified via PCR. Heterozygous mice were then bred to a separate C57BL/6J mouse line containing a FLPase deleter to remove the neomycin cassette. Mice were backcrossed for ten generations before experiments were performed. The following primers were used for determining genotype after neo-removal and back crossing; exon55_WT_F2: 5'-GCATTCTT GCTCTTTCTTGTATGG-3', DeltaN_KO_F: 5'-ACACCCAGGCAGAAGCTA GG-3', exon55_R: 5'-GAAAGGAACTCTGTCCTCTGG-3', NebEx106KI_F: 5'-AT CGTCATCTTGGCTTGGTT-3', NebEx106KI_R: 5'-TGTCTTTTTCCCTCCAA ACG-3'. Compound heterozygous NebS6366I/Δexon55 mice were produced by breeding Neb[S6366I] Het mice with Neb[Δexon55] Het mice, which had also had the neo cassette removed and were backcrossed to C57BL/6J for 10 generations. Note that each of the mutations introduced in the Neb[S6366I/Δexon55] mouse has been found in patients, but that this combination has not been detected in patients as of yet. Mice are maintained within a pathogen-free barrier facility with 14 h:10 h light: dark cycle in clear cages with access to water and food provided ad libitum. Most studies used 4 months old mice, except the studies in Fig. 1e, f (age on Figure), Fig. 8 (2 mo old), and S2 and 3 (as indicated on Figure). Male mice were used in all studies. Some studies used female mice as well but no gender differences were found in any comparisons. All animal experiments were approved by the University of Arizona and the Illinois Institute of Technology Institutional Animal Care and Use Committees and followed the NIH Guide for the Care and Use of Laboratory Animals.

**Dissection**. Mice were anesthetized with isoflurane before being euthanized by cervical dislocation. Body weights were measured and the heart, the diaphragm, and the skeletal muscles of the lower limbs were dissected and weighed; tibia length (TL) were also measured and used for normalizing muscle weights. The muscles were frozen in liquid N2.

**Grip strength**. Grip strength measurements were performed according to TREAT-NMD-protocol (www.treat-nmd.org). The mouse gripped a horizontal steel mesh with its front paws while being lifted by its tail. It was gently pulled away at a constant speed until its grip was broken. All four limbs were also tested by a similar approach; the mouse was placed on the steel mesh and gentle pulled by its tail until it released its grip. Peak tensions (grams of force) were recorded on a digital force gauge (Chatillon Force Measurement DFEII, Columbus Instruments) as the mouse released its grip.

**In-vivo muscle analysis of the gastrocnemius muscle complex**. In-vivo muscle analysis for the gastrocnemius complex was conducted using male Neb[S6366I] Hom and Compound-Het mice. Mice were anesthetized using isoflurane and placed on the heated platform (39 °C) of the Aurora Scientific Mouse Muscle Physiology System (model 809B; Aurora Scientific, Inc., Aurora, Ontario, Canada). Hair was removed from the right hind-leg and the knee immobilized using a non-invasive clamp. The foot was secured to the footplate on the force transducer (300C series with dual-mode lever systems, Aurora Scientific) with adhesive tape and set at a 90° angle. Needle electrodes were placed distal to the knee, just under the skin in close proximity to the tibial and sural nerves. Optimal needle placement and pulse phase for plantar flexion was established using 10 Hz tetanus stimulations at 40 mA. Forces were measured in mN using ASI 610A Dynamic Muscle Control v5.3 software. Optimal current was determined using twitch forces measured every 10 s. The isometric force-frequency relationship was measured at 1, 10, 20, 30, 40, 60, 80, 100, 125, 150, and 200 Hz using the same stimulation parameters as described for 10 Hz stimulations (see force-frequency sequence below). Maximal tetanic force was typically achieved at 150 Hz. Tissue weights for the gastrocnemius complex (gastrocnemius, plantaris, and soleus) were used for force normalization.

**Intact muscle mechanics**. The intact muscle mechanics have been described previously[29,59,60]. In short, EDL and soleus muscles were carefully, but quickly, excised and silk suture loops (USP 4–0) were tied to each tendon. The muscle was attached to a stationary hook and a servomotor-force transducer connected to an Aurora Scientific 1200A isolated muscle system and muscles were submerged in an oxygenated Krebs–Ringer bicarbonate solution at 30 °C (in mM: 137 NaCl, 5 KCl, 1 NaH2PO4·H2O, 24 NaHCO3, 2 CaCl2·2H2O, 1 MgSO4·7H2O and 11 glucose; pH7.4). Optimal length (L0) was found by first performing a tetanus to remove any slack in the sutures, allowing the muscle to recover, and then increasing length until twitch forces plateaued. Force-frequency relationship was determined by subjecting muscles to increasing stimulation frequencies (in Hz: 1, 10, 20, 40, 60, 80, 100 and 150 for soleus, with an additional 200 for EDL). Muscles were allowed to recover for 30, 30, 60, 90, 120, 120, 120 and 120 s between subsequent

stimulations. Force obtained (converted to mN) was normalized to the physiological CSA (PCSA) through the following equation: PCSA = mass(mg) / [muscle density(mg/mm$^3$) * fiber length(mm)]. The physiological density of muscle is 1.056 mg/mm$^3$ and fiber length was found utilizing a fiber length to muscle length ratio, 0.72 for soleus and 0.51 for EDL[61].

**Histology**. Cross sections (8 μm) of isopentane-frozen muscles were taken mid-belly and stained with modified Gomori trichrome using standard histochemical techniques. For Gomori trichrome staining, sections were stained in Harris haematoxylin (Richard Allen Scientific) for 5 min, rinsed in tap water, and then stained in Gomori trichrome solution (pH 3.4) for 10 min. Following an additional rinse and differentiation in 0.5% acetic acid for 10 s, sections were dehydrated and coverslipped using standard techniques. Light microscopic images were captured using an AxioCam MRc (Carl Zeiss, Thornwood, NY, USA).

**Transmission electron microscopy (TEM)**. For ultrastructural analysis, stretched EDL and soleus muscle were fixed in a mixture of 3.7% paraformaldehyde, 3% glutaraldehyde and 0.2% tannic acid in 10 mM PBS, pH 7.2 at 4 °C for 1 h. The muscles were then rinsed for 15 min in PBS and a postfixation was performed in 1% OsO4 in the same buffer for 30 min. Subsequently, samples were dehydrated in an ethanol graded series, infiltrated with propylene oxide and transferred to a mixture of 1:1 propylene oxide:Araldite 502/Embed 812 resin (Epon-812, EMS), then to a pure Araldite 502/Embed 812 resin, and finally polymerized for 48 h at 60 °C. Longitudinal ultrathin sections (80 nm) were obtained with a diamond knife (Diatome) in a Reichert-Jung ultramicrotome and contrasted with 1% potassium permanganate and lead citrate. Images (1792 × 1792 pixel) were acquired in a TECNAI Spirit G2 transmission electron microscope (FEI, Hillsboro, OR) with a side-mounted AMT Image Capture Engine V6.02 (4Mpix) digital camera, operated at 100 kV. ImageJ (v1.49, NIH, USA) was used for analysis of nemaline rod body size and thin filament lengths. TEM is known to cause a small amount of filament shrinkage during sample preparation (most likely during the dehydration and embedding steps) and therefore it is to be expected that the thick filaments will be slightly shorter than in vivo and to vary slightly in length from experiment to experiment. Filament shrinkage has been previously examined using the thick filament axial repeats measured on micrographs and comparing conventional transmission EM methods with quick freeze/freeze substitution methods[15]. Based on this earlier work and consistent with recent super-resolution optical microscopy[62] thick filaments can be assumed to be 1.6 μm in length. Measured thick and thin filament lengths obtained with TEM were adjusted for shrinkage accordingly.

**Super-resolution Structured Illumination Microscopy (SR-SIM)**. For SR-SIM studies WT and Compound-Het 4 mo old mice were used. Skinned EDL fiber bundles were stretched from slack at different degrees (~20–60%), embedded in O. C.T compound and immediately frozen in 2-methylbutane precooled in liquid nitrogen. 4 μm thick cryosections were then cut and mounted onto microscope slides. Tissue sections were permeabilized in 0.2% Triton X-100/PBS for 20 min at room temperature, blocked with 2% BSA and 1% normal donkey serum in PBS for 1 h at 4 °C, and incubated overnight at 4 °C with primary antibodies diluted in blocking solution. The primary antibodies included: a rabbit polyclonal anti-Tmod1 (3.33 μg/mL), rabbit polyclonal anti-ETmod (Tmod1, 3.33 μg/mL) and a mouse monoclonal anti-titin Ti102 (2.5 μg/mL). Sections were then washed with PBS for 2 × 30 min and incubated with secondary antibodies diluted in PBS for 3 h at room temperature. The secondary antibodies included: AlexaFluor-647 conjugated donkey anti-rabbit IgG (1:200, Abcam) and AlexaFluor-405 conjugated donkey anti-mouse IgG (1:200, Invitrogen). The sections were then washed with PBS for 2 × 15 min and covered with number 1.5 H coverslips (Bioscience Tools, CSHP-No1.5-24 × 60) using ProLong Diamond (Thermo Scientific, Inc.). A Zeiss ELYRA S1 SR-SIM microscope was used with UV light and solid-state laser (405/488/561/642 nm) illumination sources, a 100× oil immersion objective (NA = 1.46), and a sCMOS camera. Typical imaging was performed on a 49.34 × 49.34 μm$^2$ area with 1280 × 1280 pixel dimensions. Typical image stacks comprising of 40 slices were acquired with 0.084 μm Z-steps, five angles and five phases/angle for each slice. Image reconstruction and fluorescence intensity plot profile generation were performed with ZEN 2 software (Zeiss). Plot profiles of the antibody-labeled images were fit with Gaussian curves to determine the epitope peak position using Fityk 1.3.0 software. Thin filament length was determined from the Tmod1 epitope positions across the Z-disk. A-band width was determined from the Ti102 epitope positions across the A-band[49].

**Sample preparation and gel electrophoresis**. Muscle samples were prepared following a well-documented protocol[63]. Briefly, tissues were pulverized to powder via glass Dounce tissue Homogenizers pre-chilled in liquid nitrogen. Tissue powder was allowed to equilibrate in at −20 °C for 20 min before a 50% glycerol/H2O solution with protease inhibitors (in mM: 0.04 E-64, 0.16 leupeptin and 0.5 PMSF) and a urea buffer (in M: 8 urea, 2 thiourea, 0.050 tris–HCl, 0.075 dithiothreitol, 3% SDS w/v and 0.03% bromophenol blue, pH of 6.8) were added in a 1:40:40, sample (mg): glycerol (μL): urea (μL), ratio. The solution was mixed and incubated at 60 °C for 10 min before being aliquoted and flash frozen in liquid nitrogen.

Nebulin was visualized on 1.0% agarose gels, run at 15 mA/gel for 2 h and 50 min and stained with Coomassie blue, as described previously[63–65]. Even though no molecular weight markers for identifying nebulin exist, based on the location of the larger titin (at top of gel) and the smaller myosin (most abundant protein), nebulin can be easily identified. Its location on agarose protein gels has also been confirmed in previous work with full nebulin KO models[22]. Myosin heavy chain isoform analysis was performed as previously described[27]. Briefly, myosin heavy chain isoforms were separated using 8% acrylamide gels, run for 24 h at 15 °C and a constant voltage of 275 V and stained with Coomassie blue. Gels were scanned and analyzed with ImageJ (v1.49, NIH, USA). MHC type I and IIB are well separated on gels but the IIA overlaps with IIX due to insufficient separation. We refer to this band as IIA/X. The identity of the bands was established by running standard lysates consisting of a mixture of EDL and soleus muscles lysates, containing all isoforms in a well-established order (i.e., IIA/X at top, IIB in middle and I at bottom, e.g., see Fig. 4a, left).

**Fiber cross-sectional area analysis**. Muscle fiber cross-sectional area (CSA) was analyzed as described in Li et al.[27]. EDL and soleus muscles were pinned after stretching slightly beyond slack length on cork and covered with OCT (Tissue-Tek). The samples were then frozen in liquid nitrogen-cooled isopentane. Muscles were cut mid-belly and frozen in OCT-blocks. Six μm thick sections were collected on glass slides and stored at −20 °C for no longer than 2 days (Microm HM550, Thermo Fisher). Slides were equilibrated at room temperature for 10 min while individual sections were demarcated with a hydrophobic barrier (Vector Laboratories, Burlingame, CA, USA). Then, samples were treated with 0.2% triton X-100 in phosphate buffered saline (PBS) for 20 min followed by a 1 h incubation with a blocking solution (2% BSA, 1% normal donkey serum) in PBS at 4 °C. Primary antibodies were then applied to the sections for an overnight incubation at 4 °C: laminin (1:400 rabbit L9393, Sigma-Aldrich), MHCI (1:75 IgG2b BA-F8, DSHB), MHCIIA (1:500 IgG1 SC-71, DSHB), MHCIIX exclusion (1:100 IgG1 BF-35, DSHB) and MHCIIB (1:50 IgM BF-F3, DSHB). Following primary antibody incubation, sections were washed with PBS twice for 30 min. The matching secondary antibodies were then applied for 3–4 h at room temperature: polyclonal Alexa Fluor 488-conjugated goat anti-rabbit [1:500 IgG (H + L) A11008, Thermo Fisher], polyclonal Alexa Fluor 350-conjugated goat anti-mouse [1:500 IgG2b A211440, Thermo Fisher], polyclonal Alexa Fluor 350-conjugated goat anti-mouse [1:500 IgG1 A21120, Thermo Fisher] and polyclonal Alexa Fluor 594-conjugated goat anti-mouse [1:500 IgM (Heavy Chain) A21044, Thermo Fisher]. Postsecondary washes included two 30 min washes with PBS followed by two quick rinses with water. Images were collected using an AxioCam MRc (Carl Zeiss, Thornwood, NY, USA)[35]. CSAs were analyzed using the semi-automatic muscle analysis using segmentation of histology (SMASH ver.5) MATLAB (R2015b) application[66]. The MinFeret distance was measured since it reflects the fiber size but is relatively insensitive to cutting angle during cryosectioning[67].

**Single fiber mechanics**. EDL muscles were membrane-permeabilized (skinned) overnight at ~4 °C in relaxing solution (in mM: 40 BES, 10 EGTA, 6.56 MgCl2, 5.88 NaATP, 1 DTT, 46.35 K-propionate, 15 creatine phosphate, Ionic strength 180 mM, pH 7.0 at 20 °C) containing 1% Triton X-100 and protease inhibitors (in mM: 0.01 E64, 0.04 leupeptin and 0.5 PMSF). Muscles were then washed thoroughly with relaxing solution and stored in 50% glycerol/relaxing solution at −20 °C. Single fibers were dissected and mounted using aluminum T-clips between the length motor (ASI 322C, Aurora Scientific Inc.), and the force transducer element (ASI 403A, Aurora Scientific Inc.) in a skinned fiber apparatus (ASI 802D, Aurora Scientific Inc.). Sarcomere length was set in passive fibers to 2.5 μm using a high-speed camera and ASI 900B software (Aurora Scientific Inc. v4.196). Muscle fibers were activated in pCa (pCa = –log([Ca²⁺]) 4.0 activating solution (in mM: 40 BES, 10 CaCO3 EGTA, 6.29 MgCl2, 6.12 Na-ATP, 1 DTT, 45.3 potassium-propionate, 15 creatine phosphate, Ionic strength 180 mM, pH 7.0 at 15 °C) and protease inhibitors. Fiber width and depth (built-in prisms allow for side view of fibers and measurement of depth) were measured at four points along the fiber, and the cross-sectional area (CSA) was calculated assuming an elliptical cross-section. Specific force was expressed as force per CSA (mN/mm²).

Force-pCa curves, $k_{tr}$ and maximal shortening speed: Fibers were initially in relaxing solution and then moved into pre-activating solution (relaxing solution with a 10-fold lower EGTA concentration), followed by activation at 15 °C with incrementally increased pCa (pCa = –log([Ca2+]), ranging from 6.5 to 4.0. The obtained force–pCa relation was fitted with a Hill equation, providing $pCa_{50}$ (pCa giving 50% of maximal active force) and the Hill coefficient, nH, an index of myofilament cooperativity. Active stiffness: Active stiffness is an estimate of the number of force-generating cross-bridges during activation and was measured by lengthening and shortening the maximal activated fiber with 0.3, 0.6, 0.9% and −0.3, −0.6, −0.9% of initial fiber length. The resulting change in tension was plotted against the length change (in %) and the slope of the linear relationship is the active stiffness. $k_{tr}$-measurements: The rate of force redevelopment ($k_{tr}$) was measured at steady-state force by rapidly shortening (1 ms) the fiber by 30% at one end of the fiber resulting in unloaded shortening of the fiber for 20 ms. Remaining bound cross bridges were detached by rapidly restretching the fiber to initial length and the force redeveloped[68]. The rate constants of the double-exponential force redevelopment curve ($k_{tr}$) was determined by fitting the rise of force to the following equation (two-phase association curve): $F = Fss*((1 − e^{ktr*t}) + (1 − e^{−k*t})) + c$, where F is force at time t, Fss is steady state force, k is the constant of the slow phase of force redevelopment; $k_{tr}$ is the time constant of the fast phase of force redevelopment (the fast phase accounts for >90% of the response and $k_{tr}$ is plotted in Fig. 7). Maximal shortening velocity: The fiber was activated and allowed to reach steady state force and when rapidly shortened by 8, 9, 10, 11, 13 and 15% of initial length resulting in a brief period of unloaded shortening. The time to start of the force recovery was measured, and this time was plotted against the length change. The slope is the unloaded shortening velocity[50].

After mechanical experiments, the fiber was stored in SDS-sample buffer (15% Glycerol, 2% SDS, 62 mM Tris-HCl pH6.8, 2.5 mM Bromophenol Blue) for gel electrophoresis and myosin heavy chain isoform determination using silver-stained myosin gels (as mentioned above). Fibers containing more than 95% type IIB myosin of total myosin isoforms were considered as type IIB fibers.

**X-ray diffraction experiments**. X-ray diffraction images were recorded on intact EDL muscles with a similar approach as in Kiss et al.[30] and Ma et al.[69]. Mice were killed by cervical dislocation during carbon dioxide anesthesia and the EDL muscles were quickly excised. Silk-sutures (USP 4-0) were tied to each tendon and the muscle was mounted in a custom-built chamber filled with oxygenated solution (in mM: 145 NaCl, 2.5 KCl, 1.0 MgSO4, 1.0 CaCl2, 10.0 HEPES, 11 glucose; pH7.4) and kept at room temperature. The optimal length (L0) was determined by adjusting muscle length until maximal twitch force was produced. Force was measured with a combined motor/force transducer (Model 6350; Cambridge Technology, Bedford, MA) connected to a dual-mode controller (300C; Aurora Scientific, Aurora, Canada). Intact muscles were activated with a high-power biphasic current stimulator (model 701 A; Aurora Scientific). The whole system was controlled by an ASI 610 A data acquisition and control system (Aurora Scientific). X-ray diffraction patterns were recorded using the high-flux 12 keV X-ray beam provided by Beamline 18 at the Advanced Photon Source (Argonne National Laboratory. The BioCAT undulator beamline 18ID is a facility for biological non-crystalline diffraction and X-ray absorption spectroscopy at the Advanced Photon Source[70].). Raw image frames were collected using a Dectris Pilatus 3 1 M (Dectris AG, Baden-Dättwil, Switzerland) photon counting detector with 981 × 1043, 172 × 172 μm², pixels with a 20-bit dynamic range. Background images were recorded by passing the X-ray beam through the experimental cell just next to the muscle. For the mechanical and X-ray experiments, the muscles were activated by an 80 Hz pulse frequency, 2 ms pulse width stimulation for a 4.5 s isometric tetanic contraction. A 3 s exposure of inactive muscle was collected before activation and a 4.5 s exposure during the plateau of the tetanus with the detector continuously collecting data frames with a 10 ms exposure time and 10 ms readout time. The muscle was oscillated in the beam during the exposure to sample a large volume of the muscle and spread the X-ray dose and prevent radiation damage. After the experiment, a subset of EDL muscles was chemically fixed in 10% formalin at optimal length. Fiber bundles were dissected and their sarcomere length measured, using fiber bundles throughout the entire muscle volume.

The X-ray detector data were saved as 32-bit TIFF-images. The individual images from a muscle at rest were averaged together into a single image and the individual images taken during the tetanic force plateau were also averaged into a single image. The averaged images were analyzed using the MuscleX software (ver. 1.13.1) developed at BioCAT (https://musclex.readthedocs.io/en/latest/. J. Jiratrakanvong, J. Shao, M. Menendez, X. Li, J. Li, Weikang Ma. G. Agam, T. Irving, MuscleX: software suite for diffraction X-ray imaging V1.13.1, https://doi.org/10.5281/zenodo.1195050, March 2018.) Briefly, the lattice spacing and equatorial intensities were analyzed by the "Equator" routine in MuscleX. The images were quadrant folded and background subtracted using the "Quadrant fold" routine in MuscleX and the diffuse background intensities were summed to be used as normalization standard (details in[71]). The spacings and intensities of meridional reflections and layer lines were measured using the "Projection Trace" routine in MuscleX. The troponin and tropomyosin reflections were analyzed as described previously[30]. The thin filament helical pitches were measured by the axial spacings of the 59 Å and 51 Å actin layer lines. The 59 Å layer line (ALL6) is derived from the left-handed helix joining the G- actin subunits and the 51 Å actin layer line (ALL7) from the right-handed helix. The intensity in the layer lines were projected onto a line parallel to the meridional axis in the reciprocal radial range of $0 ≤ R ≤ 0.053$ nm$^{−1}$[72]. The axial spacings of the layer lines were estimated from the centroid of the peaks. The intensity of the 59 Å reflection was estimated by fitting the radial intensity projection of 59 Å reflection with a Gaussian function. The radial spacing of the first maxima on the 59 Å reflection in reciprocal space was estimated by calculating the distance between the centroids of the first maxima of the radially projected intensity distributions of the 59 Å layer line to the meridian. From the radial spacing (r) one can derive the distance from the center of the thin filament to the center of mass of the actin subunit (R) by assuming that the ALL6 can be modeled by a J1 Bessel function where the first maximum occurs when the Bessel function argument $2πRr$ is equal to 1.85. We define the thin filament radius to be the value of R derived from this relation.

## Statistical analysis

*General*. All measurements were obtained from distinct samples. The shown bar graphs are mean ± SEM. Statistical software used: Graphpad Prism 7. For

significant differences, P-values, T-values or F-values and Degrees of Freedom (DoF) are indicated. Symbols $*p < 0.05$; $**p < 0.01$; $***p < 0.001$; $****p < 0.0001$.

*Statistical details.* Figure 1c) Gompertz growth curves; goodness of fit: $R^2$ in WT 0.94 in Compound-Het 0.95; Test: does one curve adequately fit the data? No, $P < 0.0001$, F-value: 63.24, DoF: 344 Number of mice: WT: 21, Compound-Het: 18, and S6366I Hom: 15. d) One-way ANOVA (no matching or pairing), corrected for multiple comparisons (Tukey). Number of mice: WT: 21, Compound-Het: 18, and S6366I Hom: 15. $P = 0.004$, F-value: 6.184, DoF: 52. e) Two-way ANOVA reveals a significant effect of age ($p < 0.0001$; $F = 300$, DoF = 119) and genotype ($p < 0.0001$; $F = 19$, DoF = 119) on grip strength but without interaction. A posthoc multiple testing comparison with multiple testing correction (Dunnett) reveals a significant grip strength reduction at all 3 ages in Compound-Het mice and a significant reduction in 10 mo old NebS6366I mice. WT 18,18, 12; Compound-Het 19, 23, 14; S6366I HOM 14, 18, 6 for 1, 4, 10 mo old mice, respectively. $*p < 0.05$; $**p < 0.05$; $***p < 0.001$. f) Left: Force-frequency fit: (Y = MinForce + (MaxForce − MinForce)/(1 + (HalfFreq/X)^Hill cooef.); goodness of fit: $R^2$ in WT: 0.98, in Compound-Het: 0.97, and in S6366I Hom: 0.97; Test: does one curve adequately fit the data? No, $P < 0.0001$, F-value: 104, DoF: 216. Number of mice: WT: 9, Compound-Het: 7, and S6366I Hom: 6 mice. Middle: Force-frequency fit: (Y = MinForce + (MaxForce − MinForce)/(1 + (HalfFreq/X)^Hill coeff.); goodness of fit: $R^2$ in WT: 0.98, in Compound-Het: 0.99, and in S6366I Hom: 0.92; Test: does one curve adequately fit the data? No, $P < 0.0001$, F-value: 18, DoF: 68. Number of mice: WT: 6, Compound-Het: 2, and S6366I Hom: 6 mice. Right: Force-frequency fit: (Y = MinForce + (MaxForce − MinForce)/(1 + (HalfFreq/X)^Hill coeff.); goodness of fit: $R^2$ in WT: 0.96, in Compound-Het: 0.98, and in S6366I Hom: 0.96; Test: does one curve adequately fit the data? No, $P < 0.0001$, F-value: 27, DoF: 113. Number of mice: WT: 6, Compound-Het: 4, and S6366I Hom: 3 mice. P-values at maximal force (150 Hz) for the 3-genotypes calculated with one-way ANOVA and Dunnett's multiple comparisons test ($*p < 0.05$, $**p < 0.05$, $***p < 0.001$). The force deficit comparison in S6366I HOM mice of 3 and 10 mo was done with an unpaired two-tailed T-tests ($p = 0.04$). G) Main panel and inset: One-way ANOVA (no matching or pairing); corrected for multiple comparisons (Tukey). $P = 0.0239$, F-value: 12, DoF: 21. Number of mice: WT: 9, Compound-Het: 7, and S6366I Hom: 6 mice.

Figure 2) 2Way ANOVA with multiple comparison (multiple comparison correction (Sidak)). a) EDL; $P = 0.0003$, t-value: 4.09. DoF: 27, Plant; $P = 0.0003$, t-value: 3.22, DoF; 27, Tib Cran; $P < 0.0001$, t-value: 6.65, DoF; 27, Quad; $P < 0.0001$, t-value: 5.14, DoF; 27, Gast; $P < 0.0001$, t-value: 10.22, DoF; 27 and Diaph; $P = 0.01$, t-value: 2.74, DoF; 26. N; WT: 13; Compound-Het 16. b) Plant; $P = 0.0002$, t-value: 3.44, DoF; 21 and Gast; $P = 0.005$, t-value: 3.11, DoF; 21. N; WT: 7, S6366I Hom: 16 mice.

Figure 3) One-way ANOVA with multiple testing correction (no matching or pairing) and individual t-tests show no significant difference for any of the muscle types. Number of mice. B: WT: 9, Compound-Het: 8. Number of mice in C: WT: 7, and S6366I Hom: 7.

Figure 4a) For each muscle type: Two-way ANOVA (factors: genotype and MHC isoforms) with multiple testing correction (Sidak). F-values for interaction; EDL; $P < 0.0001$, F-value: 15.85, DoF: 39; Plant; $P < 0.0001$, F-value: 52.58, DoF; 42; Tib Cran; $P = 0.0462$, F-value: 3.31, DoF; 42; Gast; $P = 0.0007$, F-value: 8.68, DoF; 42 and Diaph; $P = 0.0009$, F-value: 8.45, DoF; 39. Number of mice. B: WT: 9, Compound-Het: 7.

Figure 4b) Two-way ANOVA (factors: genotype and fiber type) with multiple testing correction (Sidak). MinFerets for interaction effect; $P = 0.0005$, F-value: 7.49 and DoF: 37. Fiber count for interaction effect; $P = 0.0005$, F-value: 7.23 and DoF: 41. Number of mice. B: WT: 5, Compound-Het: 7 Fig. 4c) Two-way ANOVA (factors: genotype and fiber type) with multiple testing correction (Sidak). MinFerets for interaction effect; $P < 0.0001$, F-value: 22.05 and DoF: 33. Fiber count for interaction effect; $P < 0.0001$, F-value: 27.37 and DoF: 43. Number of mice. B: WT: 6, Compound-Het: 7

Figure 5b) Unpaired two-tailed T-tests. Four mice per genotype used with 5 fibers per muscle examined. Figure 5c) Unpaired two-tailed T-tests. Four mice per genotype used with 5 fibers per muscle examined. Figure 5d) One-way ANOVA (no matching or pairing); corrected for multiple comparisons (Tukey). $P < 0.0001$, F-value: 27.57, DoF: 141. Six mice per genotype used with 5 fibers per muscle examined.

Figure 6a) Force-frequency fit: (Y = MinForce + (MaxForce - MinForce)/(1 + (HalfFreq/X)^Hill cooef.); goodness of fit: $R^2$ in WT: 0.86, in Compound-Het: 0.90, and in S6366I Hom: 0.92. Test: does one curve adequately fit the data? No, $P < 0.0001$, F-value: 10, DoF: 400. b) Relative force-frequency (Y = MinForce + (MaxForce - MinForce)/(1 + (HalfFreq/X)^Hill cooef.); goodness of fit: $R^2$ in WT: 0.98, in Compound-Het: 0.98, and in S6366I Hom: 0.99; Test: does one curve adequately fit the data? No, $P < 0.0001$, F-value: 21, DoF: 400;.

Figure 6a inset and 6c–f). One-way ANOVA (no matching or pairing); corrected for multiple comparisons (Sidak). a) $P = 0.001$, F-value:8.32 and DoF:41, c) $P = 0.0017$, F-value:7.69 and DoF:38, d) $P < 0.0001$, F-value:36.3 and DoF:39, E) $P = 0.0002$, F-value:11.15 and DoF:40 and f) $P = 0.0001$, F-value:32.86 and DoF:40. Number of mice: WT: 20, Compound-Het: 11, and S6366I Hom: 11.

Figure 7a) Allosteric sigmoidal curve with as test: does one curve adequately fit the data? No, $P < 0.0001$. F-value: 8.146, DoF: 363. Goodness of fit: $R^2$ in WT: 0.60, in Compound-Het: 0.44. b–g) Unpaired two-tailed T-tests. b) $P = 0.0464$, T-value: 2.035 and DoF: 58, d) $P = 0.0282$, T-value: 2.246 and DoF: 58, On average, 7 type

IIB fibers from each of 5 WT mice and 5 type IIB fibers from each of 6 Compound-HET mice.

Figure 8c–h) Unpaired two-tailed T-tests. e) right panel $P = 0.0025$, T-value:3.3, DoF:29; f) $P < 0.0001$, F-value:5.9 DoF:31, g) $P = 0.0089$, F-value:2.954, DoF:17 and h) $P = 0.0439$, F-value:2.127, DoF:24. I) Two-way ANOVA (factors: genotype and activation) with multiple testing correction (Sidak). For genotype effect: $P < 0.0001$, F-value:48.32, DoF:65. Number of mice: WT: 15, Compound-HET: 11. For some animals both EDL muscles were used. Number of muscles: WT: 18, Compound-HET: 14.

Supplementary Fig. 2a–c) For each muscle type: Two-way ANOVA (factors: genotype and MHC isoforms) with multiple testing correction (Sidak). Number of mice. a–c) WT: 6, and S6366I Hom: 8.

Supplementary Fig. 4b) WT: Y = 0.01220*X + 1.016 (slope not significant from zero); S6366I Hom: Y = 0.008872*X + 1.043 (slope not significant from zero); Compound-Het: Y = 0.0268*X + 1.01 (slope significant from zero: $P = 0.005$, F-value:8.012, DoF:119). The slopes are not different ($p = 0.39$) but the elevation is ($p < 0.001$; F:9.6; DoF:334). 4c) One-way ANOVA (no matching or pairing); corrected for multiple comparisons (Tukey). $P < 0.0001$, F-value: 27.57, DoF: 141. Six mice per genotype used with 5 fibers per muscle examined. d left) WT: Y = 0.124*X + 0.7332 (slope $p < 0.001$); Compound-Het: Y = 0.1671*X + 0.657(slope $p < 0.001$). The slopes are not different (p:0.2674) but the elevation of the lines is different ($p < 0.001$, F29.6, DoF:291). d right) Unpaired two-tailed t-test, $p = 0.01$, F:2.11, DoF 90. e left) WT: Y = 0.07*X + 1.44 (slope $p < 0.001$); Compound-Het: Y = 0.1168*X + 1.302(slope $p < 0.001$). The slopes are different (p:0.02). e right) Unpaired two-tailed t-test, $p = 0.98$, F:1.1, DoF: 59. Number of mice: WT: 3 and Compound-Het: 3.

Supplementary Fig. 5a) Force-frequency fit: (Y = MinForce + (MaxForce − MinForce)/(1 + (HalfFreq/X)^Hill cooef.); goodness of fit: $R^2$ in WT 0.95, in Compound-Het 0.92, and in S6366I Hom 0.83. b) Relative force-frequency fit: (Y = MinForce + (MaxForce - MinForce)/(1 + (HalfFreq/X)^Hill cooef.); goodness of fit: $R^2$ in WT 0.98, in Compound-Het 0.99, and in S6366I Hom 0.99; A inset and c–f). One-way ANOVA (no matching or pairing) corrected for multiple comparisons (Sidak). c) $P = 0.0094$, F-value:5.664, DoF:27; d) $P = 0.0423$, F-value:3.047, DoF:32 and f) $P < 0.0001$, F-value:18.0, DoF:27. Number of mice: WT: 13, Compound-Het: 7, S6366I Hom: 8.

Supplementary Fig. 6a, c) Two-way ANOVA (factors: genotype and activation) with multiple testing correction (Sidak) a) For genotype effect: $P < 0.0001$, F-value:43.32, DoF:55 and c) for genotype: $P < 0.0001$, F-value:16.22, DoF:41. Number of mice: WT: 7, Compound-HET: 7. Number of muscles: WT: 10, Compound-HET: 10. b) Two-way ANOVA for 'Void', mitochondria and myofibrillar fractions analyzed on 9900x magnification images: For interaction effect: $P < 0.0001$, F-value: 13.6, DoF: 129. Number of mice: 5 WT and 5 compound Het mice were analyzed with 4 cross-sections each.

**Reporting summary**. Further information on research design is available in the Nature Research Reporting Summary linked to this article.

## Data availability

All data generated or analyzed during this study are included in this published article (and its supplementary information files). The source data underlying Figs. 1c–g, 2a, b, 3b, c, 4a–c, 5b–d, 6a–f, 7a–g, 8c–i, and Supplementary Figs. 2, 4–7 are provided as a Source Data file.

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

## Acknowledgements

We are grateful to Drs. Carina Wallgren-Pettersson and Katarina Pelin (the Folkhälsan Institute of Genetics, University of Helsinki, Finland) for advice in selecting the missense mutation. This work was supported by grants from A Foundation Building Strength, and National Institute of Arthritis and Musculoskeletal and Skin Disease grant R01AR053897. This research used resources of the Advanced Photon Source, operated for the DOE Office of Science by Argonne National Laboratory under Contract No. DE-AC02-06CH11357. This project was supported by grant 9 P41 GM103622 and 1S10OD018090-01 from the National Institute of General Medical Sciences of the National Institutes of Health.

## Author contributions

J.L., W.M, T.I. and H.G. designed the study. J.L, W.M., F.L., Y.H., J.K., B.K., P.T., R.v.d.P., E.K., H.G., J.S. and Z.H. collected the data. J.L., W.M., F.L., J.K., B.K., P.T., E.K., H.G., T.I. and H.G analyzed the data. J.L., W.M., T.I. and H.G. interpreted the data and wrote the manuscripts. All authors revised, reviewed and approved of the final version of the manuscript.

## Competing interests

The authors declare no competing interests.
