## [Peer Review File · Nature Communications]

Reviewers' Comments:

Reviewer #1:

Remarks to the Author:

The authors modified the nebulin gene in mouse and created a mouse to mimic nemaline myopathy (NM). Muscles from the mouse were examined with various techniques which show many interesting results. The paper is written very clearly.

As discussed in page 11, although the modified muscles share some features of the disease, there are many characteristics that are not seen in them, too. The authors stress the heterogeneity of the disease, but it is hard to judge if this new mouse is a better model of NM. The mechanism by which the modification employed here is affecting the muscle may not be the same as that in typical human NM. The authors are aware of this and making only modest statements but, nevertheless, it should be stressed that the link between genetic changes and the changes in muscle may not be the same.

Some of the experimental results are rather unexpected, such as lengthening of thin filaments and changes in the thin filament helical parameters. The techniques used to get these results are sound, but these observations shall be validate in further studies.

The X-ray diffraction studies showed structural features that have not been found in other studies, such as the changes in the helical twist and diameter. It is rather disappointing that these are not discussed in terms of the known structure of the thin filament. As these muscles are interesting from a structural point of view and deserve more investigation, I would expect more structural discussion. The helical pitch of actin has been known to be variable, so it is meaningful to discuss how its affect the distance between two strands.

Reviewer #2:

Remarks to the Author:

Lindqvist et al reported the first typical nemaline myopathy (NM) mouse model with compound heterozygous mutations that mimic NM patients and the results gave a novel insight for the characters of myofibrils with mutated nebulin. It is very important to understand the pathomechanism for milder NM.

Essentially, this is a well-written paper giving many works on the novel model mouse, however the following points should be considered:

1. The mutation S6366I had been reported in distal myopathy in Finnish population. Thus, this mutation makes milder phenotypes. I wonder if the authors can show chronological changes of muscle weakness (grip power and behavioral locomotion) and pathological changes (fiber type changes) on aging in their mice.
2. The nemaline bodies were observed in EM. It has been reported that the muscles in the patients with Finish distal myopathy with S6366I mutation does not display the nemaline bodies in biopsied muscles on routine staining. That will be characteristic to the muscles with missense mutation in NEB. The authors should evaluate the presence of nemaline bodies on histological examination.
3. Fig S3, the authors mention shorter thin filaments in their compound hetero mice, but in the figure S3a, the thick filaments also look shorter. The thin filament size in the example figure (S3A) were somehow shorter that those in Fig S3b. Clarify them.
4. In Fig 3A, the authors mentioned that delta-ex55 allele did not make any protein products. As a reference, the authors should show the Nebulin pattern in homozygous delta-ex55 mice.

Reviewer #1 (Remarks to the Author):

Comment: “The authors modified the nebulin gene in mouse and created a mouse to mimic nemaline myopathy (NM). Muscles from the mouse were examined with various techniques which show many interesting results. The paper is written very clearly.”

Response: Thank you for reviewing our paper and for your excellent comments that have helped to further strengthen our manuscript.

Comment: “As discussed in page 11, although the modified muscles share some features of the disease, there are many characteristics that are not seen in them, too. The authors stress the heterogeneity of the disease, but it is hard to judge if this new mouse is a better model of NM. The mechanism by which the modification employed here is affecting the muscle may not be the same as that in typical human NM. The authors are aware of this and making only modest statements but, nevertheless, it should be stressed that the link between genetic changes and the changes in muscle may not be the same.”

Response: We agree that the link between genetic changes in the Compound-Het mouse model and the muscle phenotype that we found is not necessarily the same as in patients, and we further stress this in the revised manuscript, see Discussion, page 18 towards the end of the last paragraph.

The existing nebulin KO models are not sufficient for studying therapeutic NM targets as they have phenotypes much more severe than found in patients with typical nemaline myopathy. Hence the present model is valuable in that it is based on mutations found in patients and has a phenotype that is more in line with patients that have typical NM. Importantly, the novel Compound-Het model reveals molecular mechanisms distinct from those of the severe nebulin-deficient models. This might open possibilities for therapeutic tests that could be relevant to patients.

The importance of having a range of models at our disposal for studying therapeutic targets is highlighted by our recently initiated study with the myostatin inhibitor mRK35. Previously it was shown that myostatin inhibition has no effect on muscle mass and muscle function in the conditional nebulin KO model [1]. The lack of an effect in the KO might be due to the severe disease state of this model where rescue might be very hard to achieve. However, this conclusion cannot be generalized as revealed by our study with the myostatin inhibitor mRK35 in the Compound-Het mice. In contrast to the earlier work on the Neb cKO[1] myostatin inhibition in Compound-Het mice increases muscle mass (Fig. 1, left) and improves muscle function (Fig. 1, right). This ongoing study on the Compound-Het is currently being extended, including a focus on signaling pathways. It is clear from our results so far that for therapeutic testing it is important to have the Compound-Het model of less severe nebulin-based NM in our arsenal.

Figure 1. Left) Effect of myostatin inhibition on muscle mass. In a wide range of muscle types, muscle mass is increased. Right) administering myostatin inhibitor increases grip strength to a level comparable to that of wildtype mice.

Comment: “Some of the experimental results are rather unexpected, such as lengthening of thin filaments and changes in the thin filament helical parameters. The techniques used to get these results are sound, but these observations shall be validate in further studies.”

Response: Our finding of slightly longer thin filaments in Compound-Het mice is consistent with recent work that showed that the thin filament length in skeletal muscle is not static but can lengthen through for example the action of leiomodins that give rise to a nebulin-free distal thin filament segment[2; 3; 4]. Thus the slight lengthening of the thin filament is not without precedent, and we now highlight this in our revised ms. on page 9. We are confident in the findings in the original submission but agree that additional validation studies are worthwhile. The lengthening of the thin filament was validated in two ways. First we performed additional electron microscopy (EM) studies in which we measured thin filament length on electron micrographs. Figure 2 below shows the new results overlaid with the previous ones for WT and Compound-Het and the inset compares the new and old data. These new results confirm our old findings.

Figure 2. Thin filament length in EDL muscle measured by EM. (Measurements corrected for shrinkage, see comment below.) Darker colored symbols represent results in the original manuscript (‘old’ results) and the lighter colored symbols are newly collected results. The old and new results overlap. Bar graphs shown at the bottom represent the mean \pm SEM of the data within the 2.5-3.0 μm SL range. The two data sets (‘old’ and ‘new’) are not significantly different (ANOVA with multiple testing correction (Tukey)). These new studies validate our earlier conclusion that thin filaments are slightly but significantly longer in the Compound-Het than in WT muscle. In the revised manuscript we combined the old and new results in Supplemental Figure S4b and C.

As an additional means of thin filament length validation we used super-resolution structured illumination optical microscopy (SR-SIM) and measured thin filament length in EDL muscle using the thin filament capping protein Tmod1 [2]. This also revealed slightly but significantly longer thin filaments in Compound-Het mice (See Fig. 3).

Figure 3. Thin filament length measurements in EDL muscle with SR-SIM reveals longer thin filaments in Compound-Het mice (4 mo old). Bar graphs represent the mean \pm SEM of the data within the 2.5-3.0 μm SL range. The absolute numbers are slightly different from the TEM-based values (SR-SIM-based mean TFL is \sim 20 nm longer in both genotypes) which considering the differences in techniques is not unexpected. However, both TEM and SR-SIM show significantly longer thin filaments in Compound-Hets. These new data were added to the Supplemental Figure S4d in the revised manuscript.

In **summary**, new experiments using electron microscopy and super-resolution optical microscopy (shown in Figures S4b-d of the revised ms. and discussed on page 9, middle) validate the finding of longer thin filaments in Compound-Het mice.

We also focused on validating the thin filament structural findings by performing additional X-ray diffraction studies (it is hard to get beamline at the Argonne National Laboratory but we were lucky to obtain on short notice five nightshifts for the requested validation experiments). Although our preference was to study a different muscle type (EDL muscles were studied in the original submission), X-ray diffraction experiments on intact contracting skeletal muscle have multiple requirements that include well defined tendons to attach the muscle in the X-ray rig, a small muscle cross-sectional area (to avoid hypoxia during activation) and a small fiber angle relative to the central tendon (otherwise X-ray patterns become hard to interpret).

So far only mouse EDL and soleus intact muscles have been successfully used in X-ray diffraction studies. Soleus muscle is not an ideal choice for our study because, unlike the EDL muscle, the soleus muscle has a large fiber-type switch. Hence, we decided to perform additional experiments on EDL muscles. Even though this is the same muscle type as used for our original submission, we reasoned that considering the absence of good options for studying different muscle types, studying EDL muscles in a new cohort of mice and using an X-ray rig that had been disassembled and then rebuilt is also a sound validation experiment. As shown in Fig. 4 the new results that were obtained (red symbols) overlap with the previous data (black symbols). Indeed, these new data have further increased the statistical power of our findings.

Fig. 4. Thin filament structure validation experiments. A new set of X-ray diffraction experiments was performed on intact EDL muscle (WT and Compound-HET) and new results (red symbols) were compared to existing data in the original manuscript (black symbols). a) Short-pitch helix and b) long-pitch helix of the actin filament. c) The actin radius, determined from the radial spacing of the ALL6. d) ALL6 intensity is reduced in Compound-Het mice. e) Thin-thick filament spacing in the relaxed and active states. Number of mice: Old: WT: 7, Compound-HET: 7. (For some animals both EDL muscles were used.) Number of new mice: WT: 8, Compound-HET: 4.

In **summary**, we performed additional X-ray diffraction experiments and obtained results that validated our earlier findings. We updated Figure 8 of the revised manuscript with these newly obtained results.

Comment: “The X-ray diffraction studies showed structural features that have not been found in other studies, such as the changes in the helical twist and diameter. It is rather disappointing that these are not discussed in terms of the known structure of the thin filament. As these muscles are interesting from a structural point of view and deserve more investigation, I would expect more structural discussion. The helical pitch of actin has been known to be variable, so it is meaningful to discuss how its affect the distance between two strands.”

Response: Upon re-reading our original ms. we recognize that a more extensive discussion of our X-ray diffraction findings is warranted, and we appreciate very much your comment. The ~ 59 Å and ~ 51 Å layer lines correspond to the pitches of the left-handed and right handed one-start helices of the F-actin filament, respectively (e.g., see Fig. 2A of Volkman et al [5], reproduced here and shown to the right). The

Based on Volkman et al.

The ~ 59 Å and ~ 51 Å layer line spacings are known to increase during isometric contraction and decrease during unloaded shortening, reflecting changes in thin filament strain[6]. However, unlike what one might a priori expect, the ~ 59 Å and ~ 51 Å spacings do not behave identically, neither during isometric contraction nor unloaded shortening, reflecting changes in thin filament twist [6]. Furthermore, we previously found in both passive and active muscle of the nebulin cKO mouse model that the ~ 59 Å spacing is reduced without affecting the ~ 51 Å spacing nor the baseline actin monomer axial spacing (~ 27 Å), revealing that nebulin increases the baseline thin filament helical twist[7]. The findings of the present work in the Compound-Het model (unaltered 51 Å, reduced 59 Å, no change in 27 Å actin monomer spacing) are consistent with this and support that nebulin plays a role in the twist of the F-actin filament. Furthermore, we found that the thin filament diameter is slightly but significantly increased (Fig. 8f of ms.) which is likely to reflect at least in part the increased helical twist of the actin filament.

Bordas et al.[6] developed an analytical expression for the degree of twist (and its direction) using the differential axial spacing changes of the ~ 59 Å and ~ 51 Å reflections (for detail, see[6]). Using the Bordas approach, we calculated the degree of twist, defined in terms of $\Delta n/n$ where n is the number of subunits per turn of the actin double helix, as a 0.9% increase in twist in the actin filaments in compound het EDL muscles as compared to wt. This is accomplished with no significant change in the ~ 27 Å axial repeat of the subunits (Fig. 8c of ms.) consistent with a pure rotation of the subunits around the long axis and is unlikely to require a change in the F-actin inter-strand spacing. Determining the exact molecular biophysics of the mechanisms that underlie how nebulin binding exerts these effects is an exciting challenge that we intend to address in future single molecule studies and modelling. X-ray diffraction is an averaging technique, so that while it provides very precise numbers, individual filaments may be expected to show departures from the average structure. We have made significant progress in developing modelling approaches that can assess small structural changes at the individual filament level[8] and how they may alter actomyosin interaction and force production[9], setting the stage for exciting future follow-up work.

In the revised manuscript we have added this discussion, prompted by your comment. See Discussion, bottom page 15, top page 16.

Reviewer #2 (Remarks to the Author):

Comment: “Lindqvist et al reported the first typical nemaline myopathy (NM) mouse model with Compound-Heterozygous mutations that mimic NM patients and the results gave a novel insight for the characters of myofibrils with mutated nebulin. It is very important to understand the patho-mechanism for milder NM. Essentially, this is a well-written paper giving many works on the novel model mouse, however the following points should be considered:”

Response: Thank you for reviewing our manuscript, your positive evaluation, and the excellent comments that have helped to further strengthen our manuscript.

Comment: “1. The mutation S6366I had been reported in distal myopathy in Finnish population. Thus, this mutation makes milder phenotypes. I wonder if the authors can show chronological changes of muscle weakness (grip power and behavioral locomotion) and pathological changes (fiber type changes) on aging in their mice.”

Response: In the original manuscript we had focused mainly on mice at 4 mo of age and in response to your comment we added additional age groups. We first focused on grip strength in mice 1, 4 and 10 mo of age. In Compound-Het mice grip strength was similarly reduced (~20%) in all 3 age groups (Figure 5). In the S6366I Hom mice a ~10% reduced grip strength was found in mice of 10 mo of age.

Figure 5. Grip strength in Compound-Het and NebS6366I male mice, 1, 4 and 10 mo of age. A two-way ANOVA reveals a significant effect of age and genotype on grip strength but without interaction. A posthoc multiple testing comparison with multiple testing correction (Dunnett) reveals a significant grip strength reduction at all 3 ages in Compound-Het mice and a significant reduction in 10 mo old NebS6366I mice. These new data are shown in the revised ms in Figure 1e.

As a behavioral test we individually placed mice in a round enclosure, 20 cm in diameter and 30 cm in height and then videotaped mice and scored rearing events (unsupported and supported by the enclosure wall) as well as jump events. No differences were found between genotypes in neither age group (results not shown). We do not exclude the possibility that differences would be detected with much larger cohorts of mice (we had 6-7 mice per group available). The lack of a difference in our model is however consistent with other mild myopathy models in which also no behavioral differences were detected [10] or behavioral differences were only detected after extended monitoring periods [11]. We conclude that it is possible that the strong drive of mice to explore a new environment can override a mild myopathy.

As an additional test of chronological changes of weakness we performed footplate studies (*in vivo* function of the gastrocnemius complex) in 3 and 10 mo old mice. A clear force deficit at the maximal stimulation frequency (150 Hz) was found in Compound-Hets, with a similar magnitude (~35%) in young

and old mice (Fig. 6, left and middle). Additionally we studied 10 mo old female mice that we had available and they also showed a 37% force deficit in Compound-Het mice (Fig. 6, right).

We also included in these studies S6366I Hom mice and found in both age groups and both genders a force deficit that is smaller than in Compound-Hets (Fig. 6). The average force deficit in S6366I Hom males increased in from 7% to 18% in mice 3 mo and 10 mo old, respectively ($p=0.04$).

Fig. 6. In 3 mo old male mice (left) Compound-Het mice have a 35 % decreased maximal lower limb force and NebS6366I Hom have a 7% reduction, relative to WT. In 10 mo old male mice (middle) the reduction is 37% and 18% and in 10 mo old female mice (right) the reduction is 37% and 18%. These new data are in the revised ms. Figure 1f.

Fiber type changes were studied by determining the myosin heavy chain (MHC) isoform composition at 2 and 10 months (the original ms included 4 mo old mice). We focused on soleus, EDL and TC muscles as examples of muscle types that have in Compound-Het mice large, small and no changes in fiber type composition at 4 months of age (as per our original manuscript). Fig. 7A shows that a fiber-type switch (inferred from MHC composition) towards oxidative fibers occurs in young mice and persists in old age.

In the original manuscript we had studied S6366I Hom mice at 4 mo and did not find a significant change in MHC distribution. For the revision experiments we did not have mice younger than 4 mo available but we did have a group of 10 mo old mice. This resulted in the finding of minor but significant change towards more oxidative fibers in EDL and soleus muscle of old S6366I Hom mice (Fig. 7B).

Fig. 7. Myosin heavy chain (MHC) isoform distribution in EDL, soleus and TC muscles.

A) MHC distribution at three different ages in WT and Compound-Het mice (2, 4 and 10 mo). The 4 mo old mice are from the original submission and the 2 mo and 10 mo old data were newly obtained. Results show that the fiber type switch towards oxidative fibers occurs already in the young mice (2 mo) and persists in the old mice

B) MHC distribution at two different ages in WT and S6366I Hom mice (4 mo and 10 mo). The data from the 4 mo old mice are from the original ms and the 10 mo old data are newly added. A small but significant switch towards oxidative fibers occurs in EDL and soleus muscle in the 10 mo old group. These new data are shown in the revised ms in Supplemental Figure S2b and S2c.

In **summary**, we studied chronological changes and found in Compound-Het mice a significant functional deficit and a fiber type switch that is present at young age and that persists into old age. Interestingly, we also found that in old S6366I Hom mice a phenotype appears (a deficit in grip strength, reduced *in vivo* lower limb force production and a fiber type switch) but that it smaller and occurs at a later age than in the Compound-Het mice. These new findings make an important addition to our paper and we appreciate the reviewer's comment that prompted us to do this work.

Comment: "2. The nemaline bodies were observed in EM. It has been reported that the muscles in the patients with Finish distal myopathy with S6366I mutation does not display the nemaline bodies in biopsied muscles on routine staining. That will be characteristic to the muscles with missense mutation in NEB. The authors should evaluate the presence of nemaline bodies on histological examination."

Response: Thank you for this suggestion. We have performed histology on S6366I Hom mice using Gomori trichrome staining and for comparison included studies on the Compound-Het and the conditional nebulin KO model. This revealed, as expected, that the nebulin KO muscles contain an abundance of rod bodies (Figure 8g and h), consistent with their severe structural phenotype[12]. The Compound-Het model also contains rod bodies, but they are less abundant than in the nebulin KO (Fig. 8b and e). The fewest and smallest rod bodies were detected in the S6366I Hom mice (Fig. 8c and f). The S6366I Hom histological results are overall in agreement with the study on the Finnish S6366I patients where on histology, nemaline rod bodies were either absent or only few and tiny nemaline rod bodies were observable [13]. To more precisely address correspondence between Finnish distal myopathy patients and the S6366I Hom mouse model, additional muscle types at a range of ages would have to be studied. It is also noteworthy that core-like structures were observed in ten months old Compound-Het mice (Fig 8e below). Cores have been observed in myopathy patients with nebulin mutations [14; 15]. Overall Compound-Het mice appear more severely affected than S6366I Hom mice.

Figure 8. Gomori Trichrome stained gastrocnemius sections reveal presence of nemaline rod bodies (indicated by arrows) in Compound Hets and fewer and smaller rod bodies in S6366I Homs at both 4 months (a-c) and 10 months (d-f) of age. Core-like structures were observed in 10-months old Compound-Het sections (see asterisks in e). g and h) Nemaline rod bodies in soleus (g) and EDL (h) muscles from conditional nebulin KO mice with severe nemaline myopathy as comparison. Calibration bar: 20 μ m.

In response to your comments we added to the revised manuscript text at the bottom of page 8 and top of page 9 and we added a new Figure, Supplemental Figure S3.

Comment: “3. Fig S3, the authors mention shorter thin filaments in their Compound-Hetero mice, but in the figure S3a, the thick filaments also look shorter. The thin filament size in the example figure (S3A) were somehow shorter than those in Fig S3b. Clarify them.”

Response: Transmission electron microscopy (TEM) is known to cause a small amount of filament shrinkage during sample preparation (during the dehydration and embedding steps) and therefore it is to be expected that the thick filaments look slightly shorter than 1.6 μm and to vary slightly in length from experiment to experiment (in our experiments shrinkage is typically $\sim 5\%$). This issue of filament shrinkage has been carefully examined by Hugh Huxley who used the thick filament axial repeats measured on micrographs and compared conventional transmission EM methods with quick freeze/freeze substitution methods[16]. Based on this earlier work it can be concluded that thick and thin filament lengths obtained with TEM can be adjusted for shrinkage by assuming that thick filaments are 1.6 μm in length. We have clarified the shrinkage issue in the revised manuscript, Methods section, and page 22, middle paragraph. We also provide better sample figures in Fig. S4A.

Finally we measured thick filament length with super-resolution structured illumination microscopy (SR-SIM) using the Ti102 antibody that labels the edge of the thick filaments [17] and the results reveal no difference in thick filament length between genotypes, see new Fig. S4e.

Comment: “4. In Fig 3A, the authors mentioned that delta-ex55 allele did not make any protein products. As a reference, the authors should show the Nebulin pattern in homozygous delta-ex55 mice.”

Response: We have published this model including the finding that nebulin expression levels in homozygous Δex55 mice within days after birth are reduced to $<2\%$ levels in wildtype mice [18]. In response to your comment we tested this again and confirmed that nebulin expression in skeletal muscles of homozygous Δex55 mice is undetectable, see Figure 9 below.

Figure 9. Nebulin protein expression. Nebulin expression was assessed in quadriceps muscles from WT mice and homozygous $\text{Neb}^{\Delta\text{ex55}}$ mice (5 days old). MHC functions as loading control. Nebulin is not detectable in homozygous $\text{Neb}^{\Delta\text{ex55}}$ mice.

Summary. We thank the reviewers for their comments, all of which we have taken to heart. We have performed multiple additional experiments, and added the following new Figures 1e, 1f, 5d, S2b and c, S3, S4d and S4e. We added additional data (validating earlier results) to Fig. 8, and S4a-c. The obtained results support and extend the findings of our original submission. We have also revised the text of our manuscript as indicated above. We are excited about our findings and hope that the manuscript is now ready for acceptance. Thank you again.

References.

- [1] J.A. Tinklenberg, E.M. Siebers, M.J. Beatka, B.A. Fickau, S. Ayres, H. Meng, L. Yang, P. Simpson, H.L. Granzier, and M.W. Lawlor, Myostatin Inhibition Using ActRIIB-mFc Does Not Produce Weight Gain or Strength in the Nebulin Conditional KO Mouse. *J Neuropathol Exp Neurol* 78 (2019) 130-139.
- [2] V.M. Fowler, and R. Dominguez, Tropomodulins and Leiomodins: Actin Pointed End Caps and Nucleators in Muscles. *Biophys J* 112 (2017) 1742-1760.
- [3] A. Brynneel, Y. Hernandez, B. Kiss, J. Lindqvist, M. Adler, J. Kolb, R. van der Pijl, J. Gohlke, J. Strom, J. Smith, C. Ottenheijm, and H.L. Granzier, Downsizing the molecular spring of the giant protein titin reveals that skeletal muscle titin determines passive stiffness and drives longitudinal hypertrophy. *Elife* 7 (2018).
- [4] C.T. Pappas, R.M. Mayfield, C. Henderson, N. Jamilpour, C. Cover, Z. Hernandez, K.R. Hutchinson, M. Chu, K.H. Nam, J.M. Valdez, P.K. Wong, H.L. Granzier, and C.C. Gregorio, Knockout of Lmod2 results in shorter thin filaments followed by dilated cardiomyopathy and juvenile lethality. *Proc Natl Acad Sci U S A* 112 (2015) 13573-8.
- [5] N. Volkman, D. DeRosier, P. Matsudaira, and D. Hanein, An atomic model of actin filaments cross-linked by fimbrin and its implications for bundle assembly and function. *J Cell Biol* 153 (2001) 947-56.
- [6] J. Bordas, A. Svensson, M. Rothery, J. Lowy, G.P. Diakun, and P. Boesecke, Extensibility and symmetry of actin filaments in contracting muscles. *Biophys J* 77 (1999) 3197-207.
- [7] B. Kiss, E.J. Lee, W. Ma, F.W. Li, P. Tonino, S.M. Mijailovich, T.C. Irving, and H.L. Granzier, Nebulin stiffens the thin filament and augments cross-bridge interaction in skeletal muscle. *Proc Natl Acad Sci U S A* 115 (2018) 10369-10374.
- [8] S.M. Mijailovich, M. Prodanovic, and T.C. Irving, Estimation of Forces on Actin Filaments in Living Muscle from X-ray Diffraction Patterns and Mechanical Data. *Int J Mol Sci* 20 (2019).
- [9] S.M. Mijailovich, B. Stojanovic, D. Nedic, M. Svicevic, M.A. Geeves, T.C. Irving, and H.L. Granzier, Nebulin and titin modulate cross-bridge cycling and length-dependent calcium sensitivity. *J Gen Physiol* 151 (2019) 680-704.
- [10] J. Tinklenberg, H. Meng, L. Yang, F. Liu, R.G. Hoffmann, M. Dasgupta, K.P. Allen, A.H. Beggs, E.C. Hardeman, R.S. Pearsall, R.H. Fitts, and M.W. Lawlor, Treatment with ActRIIB-mFc Produces Myofiber Growth and Improves Lifespan in the Acta1 H40Y Murine Model of Nemaline Myopathy. *Am J Pathol* 186 (2016) 1568-81.
- [11] N. Nagy, R.J. Nonneman, T. Llanga, C.F. Dial, N.V. Riddick, T. Hampton, S.S. Moy, K.K. Lehtimaki, T. Ahtoniemi, J. Puolivali, H. Windish, D. Albrecht, I. Richard, and M.L. Hirsch, Hip region muscular dystrophy and emergence of motor deficits in dysferlin-deficient Bla/J mice. *Physiol Rep* 5 (2017).
- [12] F. Li, D. Buck, J. De Winter, J. Kolb, H. Meng, C. Birch, R. Slater, Y.N. Escobar, J.E. Smith, 3rd, L. Yang, J. Konhilas, M.W. Lawlor, C. Ottenheijm, and H.L. Granzier, Nebulin deficiency in adult muscle causes sarcomere defects and muscle-type-dependent changes in trophicity: novel insights in nemaline myopathy. *Hum Mol Genet* 24 (2015) 5219-33.
- [13] C. Wallgren-Pettersson, V.L. Lehtokari, H. Kalimo, A. Paetau, E. Nuutinen, P. Hackman, C. Sewry, K. Pelin, and B. Udd, Distal myopathy caused by homozygous missense mutations in the nebulin gene. *Brain* 130 (2007) 1465-76.
- [14] C.A. Sewry, J.M. Laitila, and C. Wallgren-Pettersson, Nemaline myopathies: a current view. *J Muscle Res Cell Motil* 40 (2019) 111-126.

- [15] H. Jungbluth, S. Treves, F. Zorzato, A. Sarkozy, J. Ochala, C. Sewry, R. Phadke, M. Gautel, and F. Muntoni, Congenital myopathies: disorders of excitation-contraction coupling and muscle contraction. *Nat Rev Neurol* 14 (2018) 151-167.
- [16] H. Sosa, D. Popp, G. Ouyang, and H.E. Huxley, Ultrastructure of skeletal muscle fibers studied by a plunge quick freezing method: myofilament lengths. *Biophys J* 67 (1994) 283-92.
- [17] K. Trombitas, M. Greaser, S. Labeit, J.P. Jin, M. Kellermayer, M. Helmes, and H. Granzier, Titin extensibility in situ: entropic elasticity of permanently folded and permanently unfolded molecular segments. *J Cell Biol* 140 (1998) 853-9.
- [18] C.A. Ottenheijm, D. Buck, J.M. de Winter, C. Ferrara, N. Piroddi, C. Tesi, J.R. Jasper, F.I. Malik, H. Meng, G.J. Stienen, A.H. Beggs, S. Labeit, C. Poggesi, M.W. Lawlor, and H. Granzier, Deleting exon 55 from the nebulin gene induces severe muscle weakness in a mouse model for nemaline myopathy. *Brain* 136 (2013) 1718-31.

Reviewers' Comments:

Reviewer #1:

Remarks to the Author:

Thank you for responding to my comments adequately and thoroughly. The additional data and discussion considerably strengthened the paper. I think this is an excellent work.

Reviewer #2:

Remarks to the Author:

The authors answered my concerns well. I do not have any additional comments.